# Neural basis for regulation of vasopressin secretion by anticipated disturbances in osmolality

**Angela Kim[1,2], Joseph C Madara[1], Chen Wu[1], Mark L Andermann[1,2], Bradford B Lowell[1,2]***

[1]Division of Endocrinology, Diabetes, and Metabolism, Department of Medicine, Beth Israel Deaconess Medical Center, Harvard Medical School, Boston, United States; [2]Program in Neuroscience, Harvard Medical School, Boston, United States

**Abstract** Water balance, tracked by extracellular osmolality, is regulated by feedback and feedforward mechanisms. Feedback regulation is reactive, occurring as deviations in osmolality are *detected*. Feedforward or presystemic regulation is proactive, occurring when disturbances in osmolality are *anticipated*. Vasopressin (AVP) is a key hormone regulating water balance and is released during hyperosmolality to limit renal water excretion. AVP neurons are under feedback and feedforward regulation. Not only do they respond to disturbances in blood osmolality, but they are also rapidly suppressed and stimulated, respectively, by drinking and eating, which will ultimately decrease and increase osmolality. Here, we demonstrate that AVP neuron activity is regulated by multiple anatomically and functionally distinct neural circuits. Notably, presystemic regulation during drinking and eating are mediated by non-overlapping circuits that involve the lamina terminalis and hypothalamic arcuate nucleus, respectively. These findings reveal neural mechanisms that support differential regulation of AVP release by diverse behavioral and physiological stimuli.

**\*For correspondence:** blowell@bidmc.harvard.edu

**Competing interest:** The authors declare that no competing interests exist.

## Introduction

Rapid, anticipatory feedforward regulation is a common feature of many homeostatic circuits (*Betley et al., 2015*; *Zimmerman et al., 2016*; *Augustine et al., 2018*). While the purpose of feedforward regulation has yet to be established, a number of compelling proposals have been suggested. Several studies have shown that feedforward regulation plays an important role in shaping ingestive behavior by promoting negative-reinforcement learning (*Betley et al., 2015*; *Allen et al., 2017*; *Leib et al., 2017*). It is also suggested that feedforward regulation is crucial for maintaining energy/water balance as it prevents overconsumption by preemptively terminating ingestive behavior before systemic disturbance is detected (*Andermann and Lowell, 2017*; *Augustine et al., 2020*). Although the significance of feedforward regulation is just beginning to be recognized, the idea of such regulation in the hypothalamus existed since the late 1900 s. A conceptual framework for feedforward regulation was provided by studies looking at the effect of water intake on thirst and blood vasopressin (AVP) levels (*Rolls et al., 1980*; *Wood et al., 1980*; *Thrasher et al., 1981*; *Stricker and Hoffmann, 2007*), from which the term 'presystemic regulation' originates.

AVP, also known as antidiuretic hormone, is synthesized by posterior pituitary-projecting magnocellular AVP neurons of the paraventricular (PVH) and supraoptic nuclei (SON) of the hypothalamus. These neurons underlie the body's main non-behavioral response to dehydration or elevated systemic osmolality—an increase in circulating AVP. Upon stimulation of AVP neurons, AVP is released from the axon terminals in the posterior pituitary and enters the bloodstream (*Ferguson et al., 2003*; *Bolignano et al., 2014*). Once released, AVP binds to receptors in the kidney and stimulates water

**eLife digest** Fine-tuning the amount of water present in the body at any given time is a tight balancing act. The hormone vasopressin helps to ensure that organisms do not get too dehydrated by allowing water in the urine to be reabsorbed into the bloodstream. A group of vasopressin neurons in the brain trigger the release of the hormone if water levels get too low (as reflected by an increase in osmolality, the level of substances dissolved in a unit of blood). However, these cells also receive additional information that allows them to predict and respond to upcoming changes in water levels. For example, drinking water while dehydrated 'switches off' the neurons, even before osmolality is restored in the blood to normal levels. Eating, on the other hand, rapidly activates vasopressin neurons before the food is digested and blood osmolality increases as a result.

How vasopressin neurons receive this 'anticipatory' information remains unclear. Kim et al. explored this question in mice by inhibiting different sets of brain cells one by one, and then examining whether the neurons could still exhibit anticipatory responses. This revealed a remarkable division of labor in the neural circuits that regulate vasopressin neurons: two completely different sets of neurons from distinct areas of the brain are dedicated to relaying anticipatory information about either water or food intake.

These findings help to understand how healthy levels of water can be maintained in the body. Overall, they give a glimpse into the neural mechanisms that underlie anticipatory forms of regulation, which can also take place when hunger or thirst neurons 'foresee' that food or water will be consumed.

reabsorption to restore water balance (*Nielsen et al., 1995*). Activity of AVP neurons is regulated by two temporally distinct mechanisms. Feedback or systemic regulation, which mainly informs AVP neurons of changes in systemic water balance, is mediated by the lamina terminalis (LT), comprised of the subfornical organ (SFO), median preoptic nucleus (MnPO), and organum vasculosum lamina terminalis (OVLT) (*McKinley et al., 2004*). The SFO and OVLT are circumventricular organs (CVOs) that lack a blood-brain barrier, and are capable of sensing systemic osmolality and blood-borne factors, such as angiotensin II. Osmolality information sensed by the SFO and OVLT is then sent to AVP neurons directly or indirectly via the MnPO. Feedforward or presystemic regulation, on the other hand, informs AVP neurons of anticipated future osmotic perturbation, based on the array of presystemic signals that occur immediately before (pre-ingestive) and after (post-ingestive) food or water ingestion (*Stricker and Hoffmann, 2007*; *Stricker and Stricker, 2011*).

Presystemic regulation of AVP release was first described in 1980, in a study using dehydrated dogs (*Thrasher et al., 1981*; *Thrasher et al., 1987*), and was later replicated in humans, monkeys, sheep, and rats (*Arnauld and du Pont, 1982*; *Geelen et al., 1984*; *Blair-West et al., 1985*; *Huang et al., 2000*). These studies demonstrated that drinking causes a rapid reduction in blood AVP levels and that this decrease precedes any detectable decrease in blood osmolality (i.e., it is presystemic). However, due to lack of temporal resolution in blood AVP measurements (*Christ-Crain and Fenske, 2016*), these studies failed to capture rapid dynamics of presystemic regulation, preventing further dissection of distinct stimuli causing the presystemic reductions in AVP.

In a previous study, we demonstrated that magnocellular AVP neurons are under bidirectional presystemic regulation by drinking and eating (*Mandelblat-Cerf et al., 2017*). However, the neural inputs providing presystemic information to these neurons are still unclear. Here, we used various circuit-mapping techniques, in vivo calcium imaging, and opto- and chemo-genetics to identify and characterize the neural circuits mediating presystemic regulation of AVP neuron activity.

## Results

### AVP neurons receive inhibitory and excitatory inputs from the LT

As the presystemic regulation of AVP neurons is rapid, it must be brought about by neural afferent inputs. To identify these afferents, we performed retrograde rabies mapping from posterior pituitary-projecting, magnocellular SON[AVP] and PVH[AVP] neurons using AVP-IRES-Cre mice (*Figure 1a and e*). Cre-dependent TVA (EnvA receptor) (AAV-DIO-TVA) and rabies glycoprotein (RG) (AAV-DIO-RG)

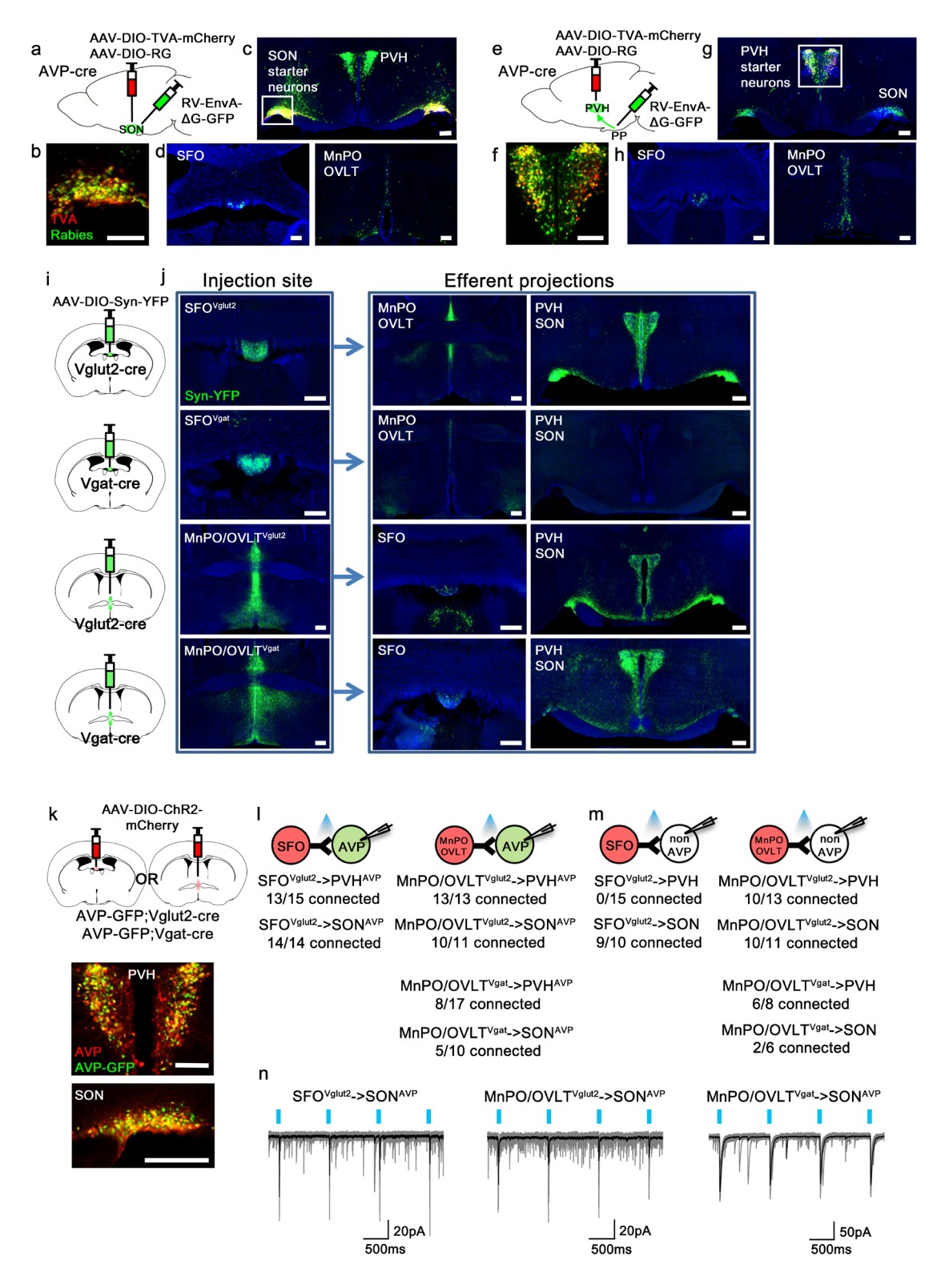

**Figure 1.** Magnocellular AVP neurons receive excitatory and inhibitory input from the LT. (**a, e**) Schematic of monosynaptic rabies tracing from magnocellular PVH[AVP] (**a**) and SON[AVP] (**e**) neurons. To target magnocellular PVH[AVP] neurons, rabies virus was injected into the posterior pituitary. (**b, f**) Representative images showing magnocellular PVH[AVP] (**b**) and SON[AVP] (**f**) starter neurons as identified by co-expression of GFP and mCherry. (**c, g**), Representative images showing magnocellular PVH[AVP] (**c**) and SON[AVP] (**g**) starter neurons and dense rabies labeling in the SON (**c**) and PVH (**g**). (**d, h**)

Figure 1 continued

Representative images showing sites containing rabies-labeled neurons in the LT that are monosynaptically connected to magnocellular PVH^AVP (d) and SON^AVP (h) neurons. (i) Schematic of anterograde tracing from excitatory and inhibitory neurons in the SFO and MnPO/OVLT. (j) Representative images showing expression of Syn-YFP in excitatory and inhibitory neurons in the SFO and MnPO/OVLT (left box), and their efferent projections in the SFO, MnPO, OVLT, PVH, and SON (right box). Note lack of YFP-labeled fibers from SFO^Vgat neurons in the PVH and SON. (k) Schematic of CRACM (top) and representative images showing co-localization of GFP and AVP immunofluorescence (red) in the PVH (middle) and SON (bottom ) of AVP-GFP mice. (l, m) Number of PVH^AVP and SON^AVP neurons (l) and non-GFP PVH and SON neurons (m) receiving direct synaptic inputs from MnPO/OVLT^Vglut2, SFO^Vglut2, and MnPO/OVLT^Vgat neurons as identified by CRACM. Mice used include *AVP-GFP;Vglut2-IRES-Cre* (MnPO/OVLT^Vglut2 and SFO^Vglut2) and *AVP-GFP;Vgat-IRES-Cre* (MnPO/OVLT^Vgat). Scale bar, 200 µm. (n) Representative traces showing light-evoked responses in SFO^Vglut2 to SON^AVP (left), MnPO/OVLT^Vglut2 to SON^AVP (middle), and MnPO/OVLT^Vgat to SON^AVP (right) CRACM. Black trace is an average of all traces (gray) in consecutive trials. AVP, vasopressin; CRACM, channelrhodopsin (ChR2)-assisted circuit mapping; LT, lamina terminalis; MnPO, median preoptic nucleus; OVLT, organum vasculosum lamina terminalis; PVH, paraventricular; SFO, subfornical organ; SON, supraoptic nuclei.

viruses were injected into the SON or PVH. After 3 weeks, EnvA-pseudotyped G-deleted rabies virus (RV-EnvA-ΔG-GFP) was injected into the SON or posterior pituitary as described below. To target magnocellular SON^AVP neurons, we injected the rabies virus directly into the SON (*Figure 1a*). Note that all AVP neurons in the SON project to the posterior pituitary (*Brown et al., 2013*; *Mandelblat-Cerf et al., 2017*). In contrast, the PVH contains magnocellular and non-posterior pituitary-projecting, parvocellular AVP neurons. To specifically target magnocellular neurons, we devised a retrograde approach in which rabies virus was injected into the posterior pituitary instead of the PVH (*Figure 1e*). Because TVA, in addition to being expressed at the cell body, is trafficked to the axon terminals (*Betley et al., 2013*; *Livneh et al., 2017*), rabies virus can be taken up by TVA-expressing axons of PVH^AVP neurons in the posterior pituitary, leading to projection-specific infection. Starter neurons in the PVH and SON were identified by co-expression of rabies-GFP and TVA-mCherry (*Figure 1b and f*). Remarkably, despite their distinctly different anatomical locations, but in agreement with their identical functions, rabies mapping results from magnocellular PVH^AVP and SON^AVP neurons were strikingly similar. Both received strong inputs from the SFO, MnPO, and OVLT (*Figure 1d and h*), and additional sparse rabies labeling was found in the arcuate nucleus (ARC) of the hypothalamus and the perinuclear zone (PNZ), an area surrounding the SON. We also found surprisingly dense rabies labeling in the PVH and SON when inputs to SON^AVP and PVH^AVP neurons were examined, respectively (*Figure 1c and g*). We performed additional experiments to investigate these unexpected findings. First, we injected AAV that Cre-independently expresses ChR2-mCherry into either the PVH or SON and found that neurons in these sites do not send direct projections to the SON and PVH, respectively. Furthermore, we also injected retrograde tracer cholera toxin subunit B into either the PVH or SON, and in agreement with the above-mentioned anterograde tracing study, failed to detect any labeled neurons in the SON and PVH, respectively (not shown). Based on these results, we believe that axosomatic and axodendritic PVH → SON^AVP or SON → PVH^AVP connections do not exist, and we speculate that the above-mentioned rabies results were caused by either non-synaptic transfer of rabies (i.e., an artifact) or synaptic transfer via axoaxonic synapses that might exist between PVH and SON neurons at the level of the posterior pituitary (*Silverman et al., 1983*; *Choudhury and Ray, 1990*).

To identify the glutamatergic and GABAergic nature of LT neurons providing inputs to PVH^AVP and SON^AVP neurons, we performed anterograde tracing from Vglut2- versus Vgat-expressing neurons of the LT. We injected the Cre-dependent synaptophysin (Syn) anterograde tracer, AAV-DIO-Syn-YFP, into the SFO and MnPO/OVLT of Vglut2-IRES-Cre and Vgat-IRES-Cre mice to individually trace glutamatergic and GABAergic projections (*Figure 1i*). The MnPO and OVLT were considered as one group because viral targeting of individual structures was challenging. We found that the MnPO/OVLT sends both excitatory and inhibitory projections to the PVH and SON while the SFO only sends excitatory projections (*Figure 1j*). Lack of long-range inhibitory projections of the SFO has been previously reported (*Oka et al., 2015*).

Next, we performed channelrhodopsin (ChR2)-assisted circuit mapping (CRACM) to test synaptic connectivity of these projections to PVH^AVP and SON^AVP neurons. AAV-DIO-ChR2-mCherry was injected into the SFO or MnPO/OVLT of AVP-GFP;Vglut2-IRES-Cre or AVP-GFP;Vgat-IRES-Cre mice and light-evoked postsynaptic currents were recorded from PVH^AVP and SON^AVP neurons identified by GFP expression (*Figure 1k*). All SON^AVP neurons recorded are magnocellular. We did not distinguish parvocellular and magnocellular PVH^AVP neurons in this experiment. We found that SFO^Vglut2 and MnPO/OVLT^Vglut2 neurons provided robust glutamatergic input to PVH^AVP and SON^AVP neurons (*Figure 1l and*

*n*). In contrast, GABAergic input from MnPO/OVLT[Vgat] neurons was detected only in half of the PVH[AVP] and SON[AVP] neurons. Next, we tested the synaptic connectivity of the LT to GFP-negative neurons in the PVH and SON (*Figure 1m*). The MnPO/OVLT provided glutamatergic and GABAergic input to the majority of GFP-negative neurons in the PVH and SON. We did not find any connection between SFO[Vglut2] neurons and GFP-negative PVH neurons, suggesting that SFO[Vglut2] inputs were highly specific to AVP neurons. Taken together, these results demonstrate that PVH[AVP] and SON[AVP] neurons receive excitatory and inhibitory inputs from the LT.

## Water-related presystemic regulation of AVP neurons is mediated by the LT

Next, we examined the involvement of the LT in presystemic regulation of SON[AVP] neurons. We focused on SON[AVP] neurons because the magnocellular population can be specifically and efficiently targeted by direct viral injection into the SON. We measured the in vivo calcium dynamics of GCaMP6s-expressing SON[AVP] neurons while inhibiting the MnPO/OVLT or SFO with CNO/hM4Di (*Figure 2a and i*). Effectiveness of hM4Di-mediated silencing of MnPO/OVLT and SFO neurons was validated in vitro in slice (*Figure 2—figure supplement 3*).

To monitor water-related presystemic responses, mice were chronically water-restricted and fully habituated to an experimental paradigm prior to the experiment. At various latencies following the beginning of each session, a water bowl was placed in the cage and mice were allowed to drink freely for ~15 min. Each mouse had one experimental session per day, in which they were pre-injected with either saline or CNO. In the control group, as previously reported (*Mandelblat-Cerf et al., 2017*), SON[AVP] neurons were rapidly inhibited as the water bowl was introduced (*Figure 2b*). This pre-ingestive response was learning-dependent as it gradually developed over the course of training (*Figure 2—figure supplement 1*). Pre- and post-ingestive suppression of SON[AVP] neurons were observed before the significant drop in systemic osmolality, which we previously found to begin 10–15 min after drinking onset (*Mandelblat-Cerf et al., 2017*). Inhibition of the MnPO/OVLT abolished the pre-ingestive drop in SON[AVP] neuron activity (*Figure 2b–f*). The post-ingestive response was also significantly attenuated. When recordings were performed from mice expressing the calcium-independent fluorescent marker, EYFP, in SON[AVP] neurons, no feeding/drinking-induced response was observed (*Figure 2—figure supplement 2*), demonstrating that the changes in GCaMP6s fluorescence were not due to movement artifacts.

Because the MnPO contains thirst-regulating neurons (*Abbott et al., 2016*; *Augustine et al., 2018*), we also analyzed drinking behavior in these mice. MnPO/OVLT inhibition did not prevent drinking but delayed drinking onset as demonstrated by a significant increase in the latency to drink (*Figure 2g*). The number of drinking bouts per session was not affected. The MnPO/OVLT, along with the SFO, is the main source of excitatory input that underlies activation of SON[AVP] neurons during systemic hyperosmolality (*McKinley et al., 2004*). Consistent with this, we found that MnPO/OVLT inhibition caused a drop in the baseline activity of SON[AVP] neurons in water-restricted mice (*Figure 2h*). In a separate group of mice, we investigated the effect of SFO inhibition (*Figure 2i*). Inhibition of the SFO caused a slight but non-significant attenuation of pre- and post-ingestive suppression (*Figure 2j–l*). Taken together, these data indicate that neurons in the MnPO/OVLT are important mediators of water-related presystemic signals to AVP neurons.

## Presystemic signals enter the LT at the level of MnPO

Next, we sought to understand the organization of the presystemic circuit mediating water-related information within and upstream of the LT. Both the SFO and MnPO/OVLT provide strong excitatory inputs to AVP neurons. However, we found that only the MnPO/OVLT was required for water-related presystemic regulation of SON[AVP] neurons (*Figure 2*), suggesting that the presystemic circuit might be organized similarly to the hierarchical thirst circuit described previously (*Augustine et al., 2018*). To test this idea, we mapped inputs to SON-projecting MnPO/OVLT[Vglut2] and SFO[Vglut2] neurons using monosynaptic rabies tracing. SON-projecting neurons were targeted using a retrograde approach shown schematically in *Figure 3a and d*. SON-projecting starter neurons, visualized by co-expression of GFP and mCherry, were seen in the MnPO/OVLT and SFO (*Figure 3b and e*). Results from the MnPO/OVLT and SFO were strikingly different. SON-projecting MnPO/OVLT[Vglut2] neurons received inputs from multiple sites (*Figure 3c*), including the SFO, PVH, ventromedial (VMH), and dorsomedial

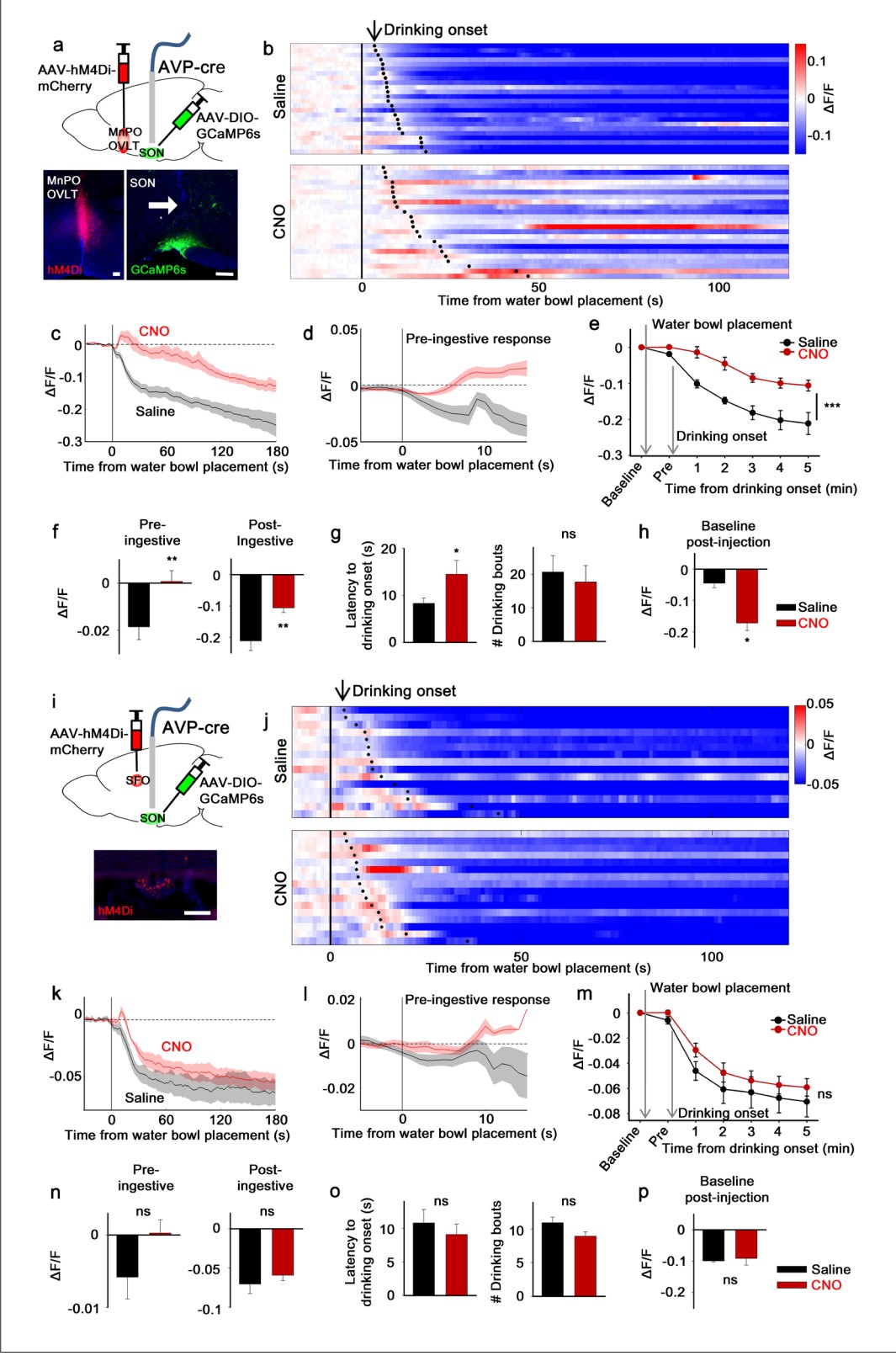

**Figure 2.** The LT mediates water-related presystemic regulation of SON[AVP] neurons. (**a, i**) Schematic of SON[AVP] photometry experiment with hM4Di-mediated non-specific inhibition of neurons in the MnPO/OVLT (**a**) and SFO (**i**). (**b, j**) Single-trial timecourses of SON[AVP] population activity in response to water bowl placement in saline and CNO trials of mice expressing hM4Di in the MnPO/OVLT (**b**) and SFO (**j**). Trials are sorted according to latency from

*Figure 2 continued on next page*

*Figure 2 continued*

water bowl placement to drinking onset (black ticks). n=6 (MnPO), 5 (SFO) mice. (**c, k**), Average SON$^{AVP}$ population activity in response to water bowl placement in saline and CNO trials of mice expressing hM4Di in the MnPO/OVLT (**c**) and SFO (**k**). n=6 (MnPO), 5 (SFO) mice. (**d, l**) Average of pre-ingestive responses in saline and CNO trials of mice expressing hM4Di in the MnPO/OVLT (**d**) and SFO (**l**). n=6 (MnPO), 5 (SFO) mice. Values are means ± SEMs across trials. (**e, m**) Data from panels (c) (**e**) and (k) (**m**) binned across drinking periods. ***p<0.001; repeated measures (RM) two-way ANOVA, p<0.001 (MnPO), p>0.005 (SFO), n=6 (MnPO), 5 (SFO) mice. (**f, n**) Average pre- and post-ingestive responses in saline and CNO trials of mice expressing hM4Di in the MnPO/OVLT (**f**) and SFO (**n**). ns, p>0.05; *p<0.05; **p<0.01; paired t-test, p=0.00974, 0.00414 (MnPO), 0.101, 0.361 (SFO), n=6 (MnPO), 5 (SFO) mice. (**g, o**) Average latency to drinking onset and number of drinking bouts in saline and CNO trials of mice expressing hM4Di in the MnPO/OVLT (**g**) and SFO (**o**). ns, p>0.05; *p<0.05; **p<0.01; paired t-test, p=0.197, 0.0302 (MnPO), 0.0888, 0.141 (SFO), n=6 (MnPO), 5 (SFO) mice. (**h, p**) Average change in baseline activity in saline and CNO trials of mice expressing hM4Di in the MnPO/OVLT (**h**) and SFO (**p**). ns, p>0.05; *p<0.05; **p<0.01; paired t-test, p=0.0152 (MnPO), 0.692 (SFO), n=6 (MnPO), 3 (SFO) mice. Values are means ± SEMs across mice except for (**d, l**). See also *Figure 2—figure supplements 1–3*, AVP, vasopressin; LT, lamina terminalis; MnPO, median preoptic nucleus; OVLT, organum vasculosum lamina terminalis; PVH, paraventricular; SFO, subfornical organ; SON, supraoptic nuclei.

The online version of this article includes the following source data and figure supplement(s) for figure 2:

**Source data 1.** SON$^{AVP}$ neuron response to water bowl placement after non-specific inhibition of the MnPO/OVLT and SFO.

**Figure supplement 1.** Pre-ingestive inhibition of SON$^{AVP}$ neurons by water cue develops gradually over training.

**Figure supplement 1—source data 1.** Response of SON$^{AVP}$ neurons to water bowl placement during training.

**Figure supplement 2.** Lack of water- and food-induced responses in EYFP-expressing AVP-IRES-Cre mice.

**Figure supplement 2—source data 1.** Fluorescence changes in the SON of EYFP-expressing AVP-IRES-Cre mice in response to water and food bowl placement.

**Figure supplement 3.** Effect of CNO application on hM4Di-expressing neurons of the MnPO/OVLT and the SFO.

**Figure supplement 3—source data 1.** Effect of CNO application on hM4Di-expressing neurons of the MnPO/OVLT and the SFO.

---

nuclei (DMH) of the hypothalamus, and the lateral parabrachial nucleus (LPBN). Among these areas, the SFO contained the greatest number of rabies-labeled neurons, demonstrating strong interconnectivity between these LT structures. In contrast, SON-projecting SFO$^{Vglut2}$ neurons received inputs only from the other LT structures, the MnPO and OVLT (*Figure 3f*). The result was further validated using Nos-1-IRES-Cre mice, another Cre line that specifically marks excitatory SFO neurons (*Figure 3— figure supplement 1*; *Zimmerman et al., 2016*). These data suggest that the MnPO/OVLT is the primary LT region that receives and relays extra-LT inputs to AVP neurons and, thus, is likely the main entry point for water-related presystemic information.

## Collateralization of SON-projecting LT neurons

Magnocellular PVH$^{AVP}$ and SON$^{AVP}$ neurons, despite their distinct anatomical locations, are functionally identical, receive inputs from the same sites (*Figure 1*), and show similar presystemic responses (*Mandelblat-Cerf et al., 2017*). To explore the possibility that presystemic regulation of both PVH$^{AVP}$ and SON$^{AVP}$ neurons is mediated by collateral projections from the same LT neurons, we performed rabies-based axon collateral mapping from SON-projecting MnPO/OVLT$^{Vglut2}$, MnPO/OVLT$^{Vgat}$, and SFO$^{Vglut2}$ neurons and examined their collateral projections in the LT, PVH, and SON. Rabies-based axon collateral mapping was performed as previously described (*Betley et al., 2013*; *Livneh et al., 2017*; *Livneh et al., 2020*). We found rabies-labeled collateral projections of SON-projecting MnPO/ OVLT$^{Vglut2}$, MnPO/OVLT$^{Vgat}$, and SFO$^{Vglut2}$ neurons in the PVH and LT (*Figure 3g–i*). The density of collateral projections was lower than that of the projections to the SON, suggesting that only a subset of SON-projecting MnPO/OVLT$^{Vglut2}$, MnPO/OVLT$^{Vgat}$, and SFO$^{Vglut2}$ neurons send collaterals to the PVH and LT. Interestingly, we found that MnPO/OVLT-projecting SFO$^{Vglut2}$ neurons, which include thirst-promoting neurons (*Matsuda et al., 2017*; *Augustine et al., 2018*), send a similar density of rabies-labeled fibers to the MnPO/OVLT, PVH, and SON (*Figure 3j*). This result suggests that a majority of these SON-projecting SFO$^{Vglut2}$ neurons send collaterals to the MnPO/OVLT and PVH, and, therefore may be capable of simultaneously regulating AVP release and thirst. Taken together, these results

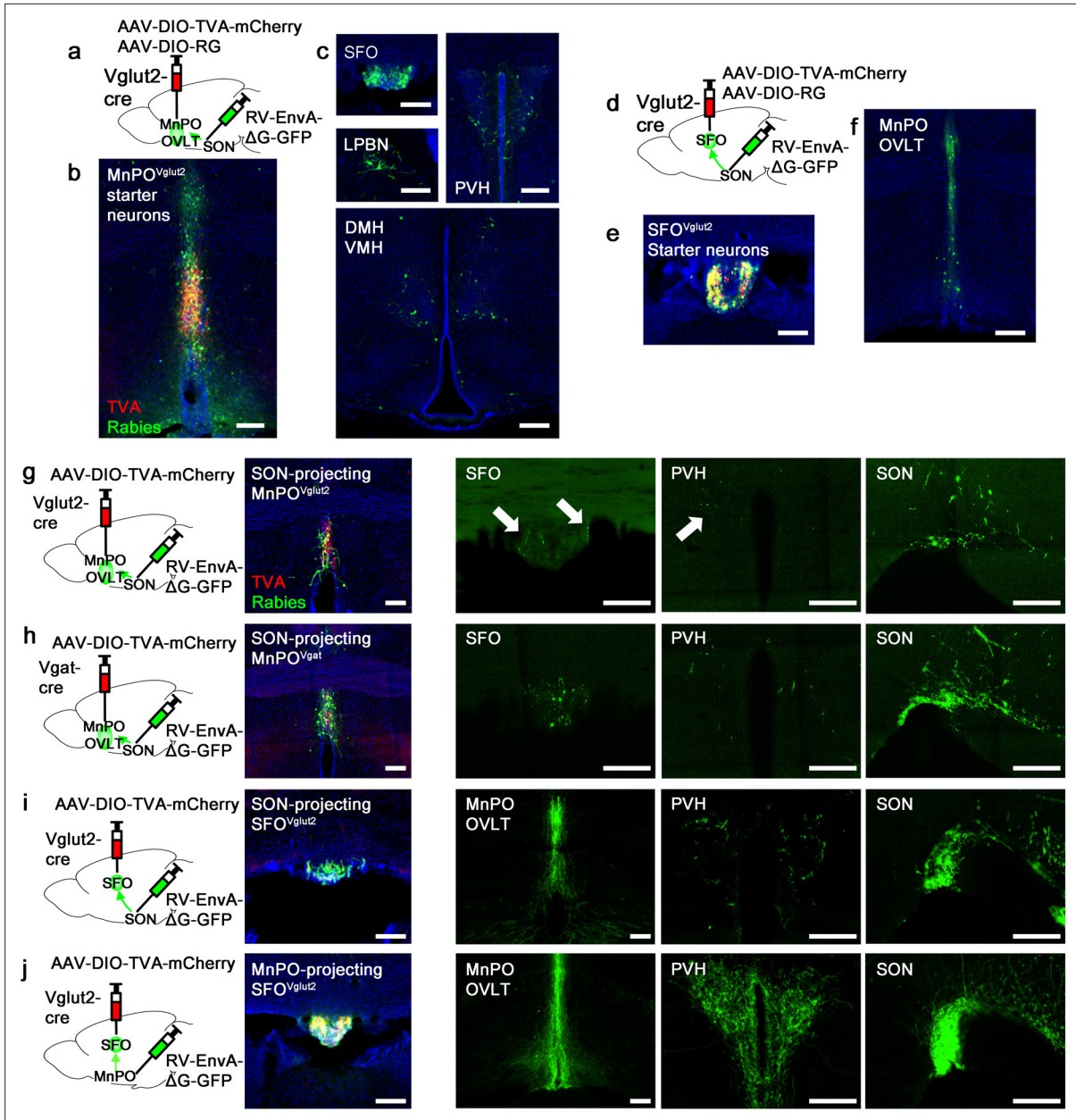

**Figure 3.** Organization of water-related presystemic neural circuit. (**a**) Schematic of monosynaptic rabies tracing from MnPO/OVLT$^{Vglut2}$ neurons. (**b, c**) Representative images showing MnPO/OVLT$^{Vglut2}$ starter neurons as identified by co-expression of GFP and mCherry (**b**), and sites containing rabies-labeled neurons that are monosynaptically connected to MnPO/OVLT$^{Vglut2}$ neurons (**c**). (**d**) Schematic of monosynaptic rabies tracing from SFO$^{Vglut2}$ neurons. (**e, f**) Representative image showing SFO$^{Vglut2}$ starter neurons as identified by co-expression of GFP and mCherry (**e**), and sites containing rabies-labeled neurons that are monosynaptically connected to SFO$^{Vglut2}$ neurons (**f**). (**g–j**) Schematic of rabies-based axon collateral mapping (left), representative images showing starter neurons (middle), and rabies-labeled collateral projections (right) of SON-projecting MnPO/OVLT$^{Vglut2}$ (**g**), MnPO/OVLT$^{Vgat}$ (**h**), and SFO$^{Vglut2}$ (**i**) neurons and MnPO/OVLT-projecting SFO$^{Vglut2}$ (**j**) neurons. Arrows, rabies-labeled collateral projections. Scale bar, 200 μm. See also *Figure 3—figure supplement 1*. MnPO, median preoptic nucleus; OVLT, organum vasculosum lamina terminalis; SFO, subfornical organ; SON, supraoptic nuclei.

The online version of this article includes the following figure supplement(s) for figure 3:

**Figure supplement 1.** Monosynaptic rabies tracing from SON-projecting SFO$^{Nos-1}$ neurons showing lack of extra LT afferents.

indicate that, unlike convergence of afferent presystemic inputs to the MnPO/OVLT, downstream projections of the LT are redundantly organized such that presystemic information is shared not only between the LT structures but also between PVH$^{AVP}$ and SON$^{AVP}$ neurons via collateral projections.

## SON-projecting LT neurons show presystemic responses to water-predicting cues and drinking

Next, we measured in vivo calcium dynamics of SON-projecting MnPO/OVLT and SFO neurons in response to water bowl presentation and drinking. GCaMP6s was expressed in SON-projecting excitatory or inhibitory neurons of the MnPO/OVLT and SFO using a modified herpes simplex virus that Cre-dependently expresses GCaMP6s (HSV-GCaMP6s) and retrogradely infects neurons from the axon terminals (*Kohl et al., 2018*). HSV-GCaMP6s was injected into the SON of Vglut2-IRES-Cre or Vgat-IRES-Cre mice and an optic fiber was placed above the MnPO/OVLT or SFO (*Figure 4a and g*). We found that SON-projecting MnPO/OVLT$^{Vglut2}$ and SFO$^{Vglut2}$ neurons both showed an immediate drop in activity upon water bowl placement (*Figure 4b–e*). A further drop in activity was observed as drinking continued. This resembled the response of SON$^{AVP}$ neurons, suggesting that the inputs from MnPO/OVLT$^{Vglut2}$ and SFO$^{Vglut2}$ neurons contribute to water-related presystemic regulation of AVP neurons. MnPO/OVLT$^{Vglut2}$ neurons showed a larger and sharper decrease in activity compared to SFO$^{Vglut2}$ neurons when normalized values were compared (*Figure 4f*). Activity of MnPO/OVLT$^{Vglut2}$ neurons reached <50% of baseline activity within the first 1 min of drinking, whereas SFO$^{Vglut2}$ neurons showed a gradual decrease that continued for over 5 min. A sharper drop in MnPO/OVLT$^{Vglut2}$ neuron activity is consistent with significant attenuation of water-related presystemic response by MnPO, but not SFO, inhibition.

The SON-projecting MnPO/OVLT$^{Vgat}$ neurons showed a completely different response (*Figure 4h–k*). Placement of the water bowl caused an abrupt increase in MnPO/OVLT$^{Vgat}$ neuron activity, which rapidly declined and returned to baseline within 1 min. No drinking-related, long-term responses were observed. We hypothesize that SON-projecting MnPO/OVLT$^{Vgat}$ neurons mainly signal the anticipation of water and work in parallel with MnPO/OVLT$^{Vglut2}$ and SFO$^{Vglut2}$ neurons to ensure rapid inhibition of AVP neuron activity upon water bowl placement. Taken together, these data indicate that the MnPO/OVLT is a key relay center for water-related presystemic information.

## Excitatory and inhibitory neurons of the MnPO/OVLT contribute to different aspects of water-related presystemic response of AVP neurons

As discussed earlier, SON-projecting excitatory and inhibitory neurons of the LT showed distinct patterns of presystemic response to water cue and drinking, suggesting that presystemic information conveyed by these neurons to AVP neurons may differ. To test this idea, we selectively inhibited excitatory and inhibitory MnPO/OVLT neurons with CNO/hM4Di while monitoring in vivo calcium dynamics of SON$^{AVP}$ neurons in response to water bowl presentation and drinking (*Figure 5a and j*). This was achieved by injecting a flp-dependent hM4Di virus (AAV-fDIO-hM4Di-mCherry) into the MnPO/OVLT of AVP-IRES-Cre;Vglut2/Vgat-IRES-Flp mice. Electrophysiological recordings demonstrated that CNO/hM4Di effectively silenced both excitatory and inhibitory MnPO/OVLT neurons in Vglut2- and Vgat-IRES-Flp mice, respectively (*Figure 2—figure supplement 3*).

Inhibition of MnPO/OVLT$^{Vglut2}$ neurons caused an overall reduction of water-related presystemic responses in SON$^{AVP}$ neurons. Both pre- and post-ingestive responses were significantly affected (*Figure 5b–f*). However, in comparison to non-specific silencing of the same area, which caused a complete removal of the pre-ingestive response of SON$^{AVP}$ neurons (*Figure 2f*), inhibition of excitatory neurons in the MnPO/OVLT was less effective in reducing pre-ingestive component of the response as ~30 % of the pre-ingestive response persisted (*Figure 5g*). Post-ingestive responses (*Figures 2f and 5g*) and baseline activity before water bowl presentation (*Figures 2h and 5i*) were reduced to a similar degree in both cases. No significant change in latency to drink or in the number of drinking bouts was observed during the experimental session. However, when mice were tested in a separate setup using a lickometer, MnPO/OVLT$^{Vglut2}$ neuron inhibition caused a significant reduction in the amount of water intake in the same group of animals (*Figure 5h*).

In contrast, the effect of MnPO/OVLT$^{Vgat}$ neuron inhibition was strictly restricted to the pre-ingestive phase (*Figure 5k–o*). More specifically, a significant reduction in the SON$^{AVP}$ neuron response was

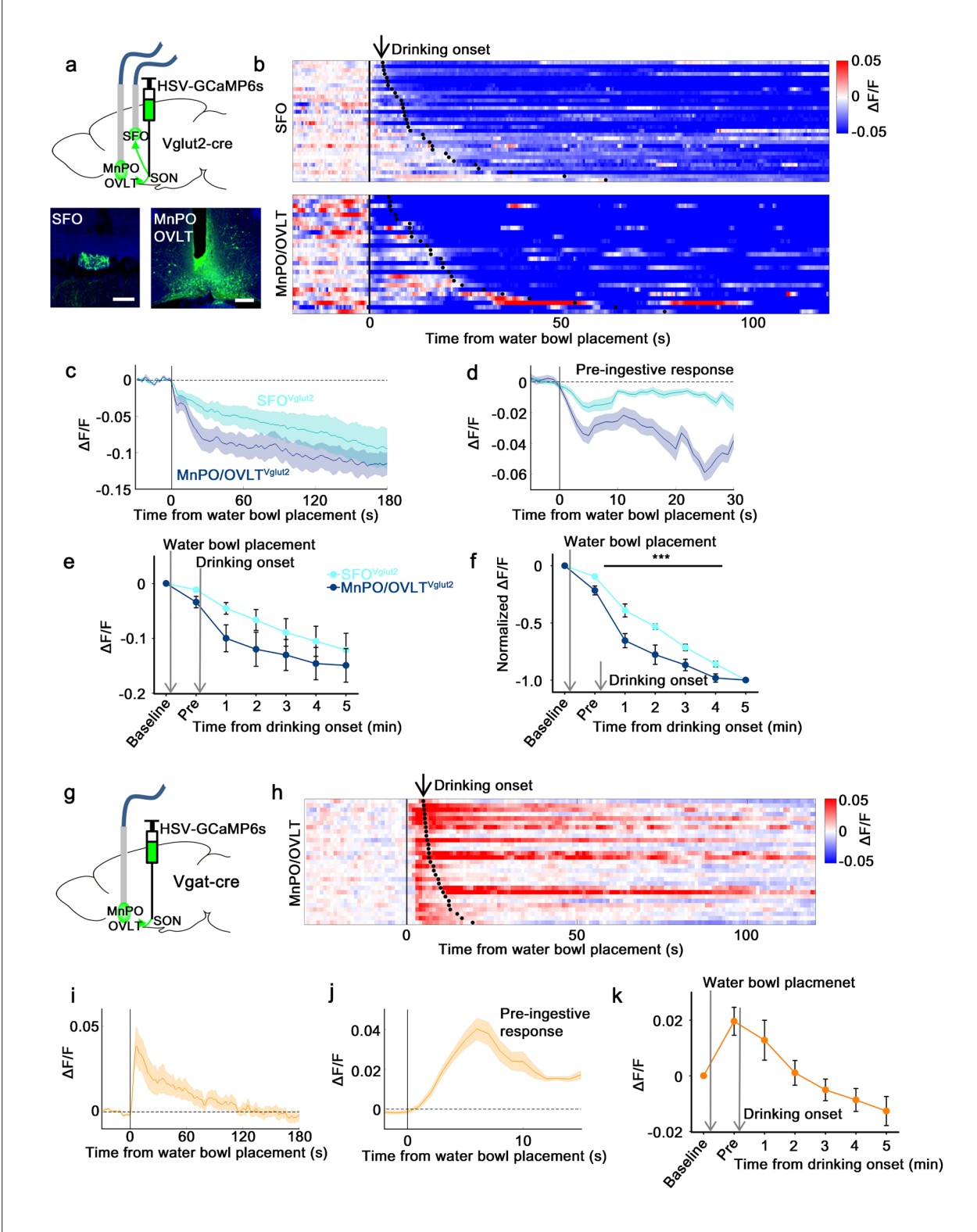

**Figure 4.** SON-projecting SFO[Vglut2], MnPO/OVLT[Vglut2], and MnPO/OVLT[Vgat] neurons show presystemic responses to water bowl placement and drinking. (**a**) Schematic of photometry experiment from SON-projecting SFO[Vglut2] and MnPO/OVLT[Vglut2] neurons. (**b**) Single-trial timecourses of SON-projecting SFO[Vglut2] (top) and MnPO/OVLT[Vglut2] (bottom) population activity in response to water bowl placement. Trials are sorted according to latency from water bowl placement to drinking onset (black ticks). n=7 (SFO[Vglut2]), and 5 (MnPO/OVLT[Vglut2]) mice. (**c**) Average population activity of SON-projecting SFO[Vglut2] (light blue) and MnPO/OVLT[Vglut2] (dark blue) neurons in response to water bowl placement. n=7 (SFO[Vglut2]), and 5 (MnPO/OVLT[Vglut2]) mice. (**d**) Average

*Figure 4 continued on next page*

*Figure 4 continued*

pre-ingestive responses of SON-projecting SFO$^{Vglut2}$ (light blue) and MnPO/OVLT$^{Vglut2}$ (dark blue) neurons. n=7 (SFO$^{Vglut2}$), and 5 (MnPO/OVLT$^{Vglut2}$) mice. Values are means ± SEMs across trials. (**e**) Data from panel (**c**) binned across drinking periods. n=7 (SFO$^{Vglut2}$), and 5 (MnPO/OVLT$^{Vglut2}$) mice. (**f**) Normalized average SON-projecting SFO$^{Vglut2}$ and MnPO/OVLT$^{Vglut2}$ population activity binned across drinking periods. Values are normalized to the total change. ***p<0.001; two-way ANOVA, p<0.001, n=7 (SFO$^{Vglut2}$), and 5 (MnPO/OVLT$^{Vglut2}$) mice. (**g**) Schematic of photometry experiment from SON-projecting MnPO/OVLT$^{Vgat}$ neurons. (**h**) Single-trial timecourses of SON-projecting MnPO/OVLT$^{Vgat}$ population activity in response to water bowl placement. Trials are sorted according to latency from water bowl placement to drinking onset (black ticks). n=9 mice. (**i**) Average population activity of SON-projecting MnPO/OVLT$^{Vgat}$ neurons in response to water bowl placement. n=9 mice. (**j**) Average pre-ingestive responses of SON-projecting MnPO/OVLT$^{Vgat}$ neurons. n=9 mice. Values are means ± SEMs across trials. (**k**) Data from panel (**i**) binned across drinking periods. n=9 mice. Scale bar, 200 μm. Values are means ± SEMs across mice except for (**d, j**). MnPO, median preoptic nucleus; OVLT, organum vasculosum lamina terminalis; SFO, subfornical organ; SON, supraoptic nuclei.

found only during the last 5 s of pre-ingestive period that immediately precedes the onset of drinking (*Figure 5p*). This result is in line with our previous finding that MnPO/OVLT$^{Vgat}$ neurons show a transient activation to water-predicting cues (*Figure 4h–k*). The degree of reduction in pre-ingestive responses (*Figure 5p*) was comparable to that of non-specific silencing of the MnPO/OVLT (*Figure 2f*), suggesting that water-predicting cue/pre-ingestive information is primarily conveyed by inhibitory neurons of the MnPO/OVLT. Consistent with a lack of post-ingestive or any sustained changes in the activity of MnPO/OVLT$^{Vgat}$ neurons in response to drinking (*Figure 4h–k*), their inhibition had no effect on post-ingestive responses or baseline activity of SON$^{AVP}$ neurons (*Figure 5p and r*). Drinking behavior was not altered during the experimental session or when tested in a separate setup (*Figure 5q*). Taken together, these results demonstrate that excitatory and inhibitory inputs from the MnPO/OVLT are each responsible for pre- and post-ingestive drops in SON$^{AVP}$ neuron activity, respectively, and that both inputs are required for water-related presystemic responses of SON$^{AVP}$ neurons.

## The LT does not mediate food-related presystemic regulation of AVP neurons

We found that the LT is involved in water-related presystemic regulation of SON$^{AVP}$ neurons with the MnPO/OVLT playing a key role. We next tested whether the same region was involved in food-related presystemic regulation using the same approach as shown in *Figure 2a*. Mice were chronically food-restricted and were trained to take food pellets from a food bowl placed in the cage with varying latency. In the control group, as previously reported (*Mandelblat-Cerf et al., 2017*), SON$^{AVP}$ neurons showed an increase in activity that began immediately upon feeding onset (*Figure 6a*). MnPO/OVLT inhibition did not affect the baseline activity (not shown) or feeding-induced activation (*Figure 6a–c*). These results demonstrate that food-related presystemic regulation is mediated by a different circuit that does not involve the MnPO/OVLT.

We then explored the activity of SON-projecting LT neurons during food intake. In line with the result from MnPO/OVLT inhibition, none of the SON-projecting LT neurons displayed responses that could account for feeding-induced activation observed in SON$^{AVP}$ neurons (*Figure 6d and e*). SON-projecting SFO$^{Vglut2}$ and MnPO/OVLT$^{Vglut2}$ neurons showed an overall decrease in activity after food presentation but no consistent pattern was observed. SON-projecting MnPO/OVLT$^{Vgat}$ neurons were activated by food presentation but their activity gradually declined following feeding onset. The amplitude of these responses was significantly smaller than water-related responses of the same neurons (*Figure 6e*).

Presystemic activation of SON$^{AVP}$ neurons is time-locked to the feeding onset (this study and *Mandelblat-Cerf et al., 2017*). To better compare responses in the LT neuron responses and in SON$^{AVP}$ neurons, we reanalyzed the data by aligning the traces to feeding onset. SON-projecting SFO$^{Vglut2}$, MnPO/OVLT$^{Vglut2}$, and MnPO/OVLT$^{Vgat}$ neurons all showed a decrease in activity (*Figure 6f*). Even if these were to be bona fide responses, a small, simultaneous decrease in excitatory and inhibitory tone is unlikely to be the primary cause of rapid feeding-induced activation of SON$^{AVP}$ neurons.

A previous study demonstrated rapid net activation of all excitatory SFO neurons by food intake (*Zimmerman et al., 2016*), which contradicts our finding. To resolve this inconsistency, we performed fiber photometry from the entire SFO$^{Vglut2}$ population. When compared to SON$^{AVP}$ neurons, the response of SFO$^{Vglut2}$ neurons was significantly delayed and lacked an early post-ingestive component occurring within 1 min of feeding onset (*Figure 6g*). Given that blood osmolality begins to

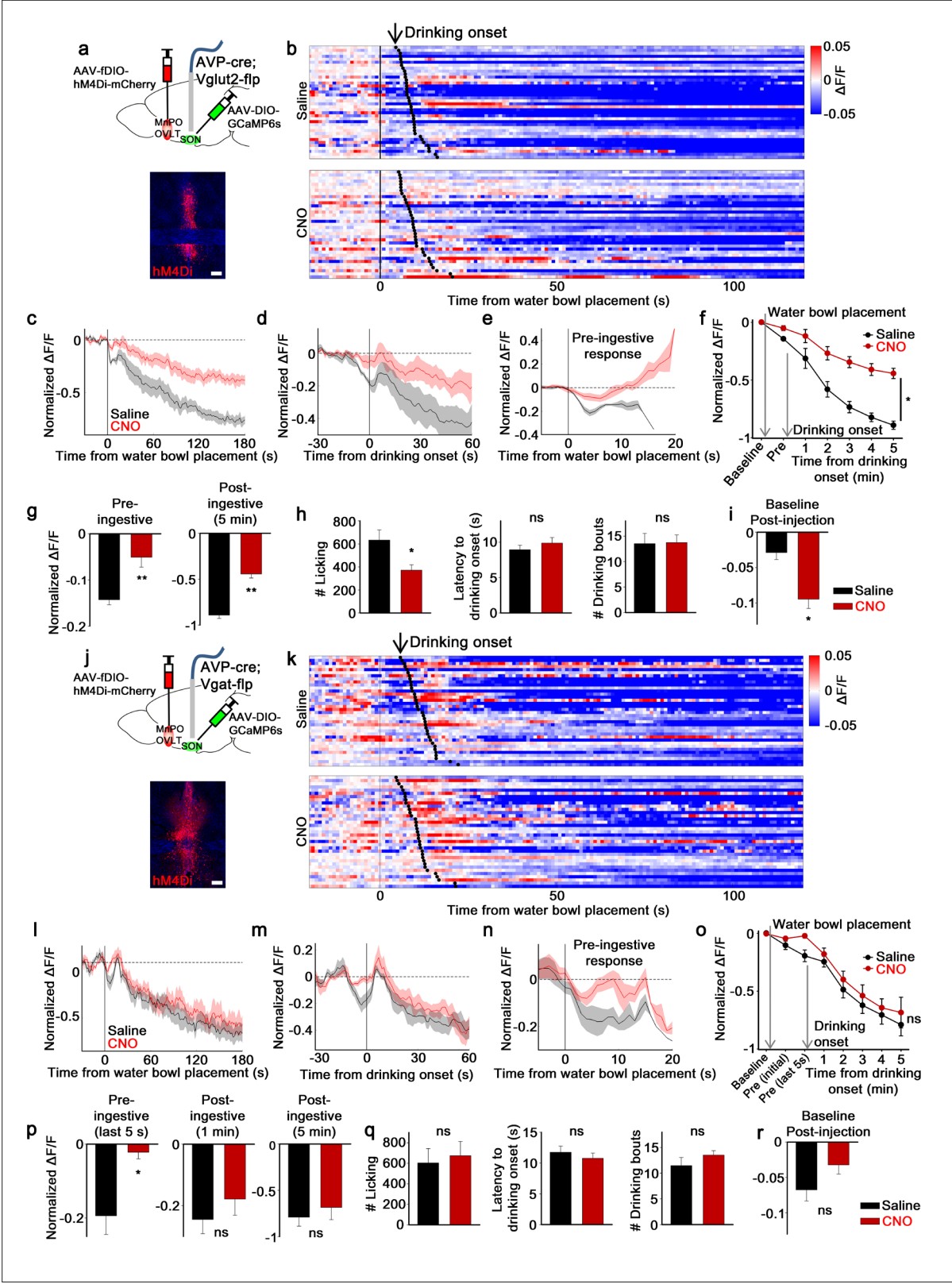

**Figure 5.** MnPO/OVLT[Vglut2] and MnPO/OVLT[Vgat] neurons contribute to different aspects of water-related presystemic regulation of SON[AVP] neurons. (**a**, **j**) Schematic of SON[AVP] photometry experiment with hM4Di-mediated inhibition of MnPO/OVLT[Vglut2] (**a**) and MnPO/OVLT[Vgat] (**j**) neurons. (**b**, **k**) Single-trial timecourses of SON[AVP] population activity in response to water bowl placement in saline and CNO trials of mice expressing hM4Di in MnPO/OVLT[Vglut2] (**b**) and MnPO/OVLT[Vgat] (**k**) neurons. Trials are sorted according to latency from water bowl placement to drinking onset (black ticks). n=6

*Figure 5 continued on next page*

Figure 5 continued

(MnPO/OVLT$^{Vglut2}$), 6 (MnPO/OVLT$^{Vgat}$) mice. (**c, l**) Average SON$^{AVP}$ population activity in response to water bowl placement in saline and CNO trials of mice expressing hM4Di in MnPO/OVLT$^{Vglut2}$ (**c**) and MnPO/OVLT$^{Vgat}$ (**l**) neurons. n=6 (MnPO/OVLT$^{Vglut2}$), 6 (MnPO/OVLT$^{Vgat}$) mice. (**d, m**) Average SON$^{AVP}$ population activity in response to drinking onset in saline and CNO trials of mice expressing hM4Di in MnPO/OVLT$^{Vglut2}$ (**d**) and MnPO/OVLT$^{Vgat}$ (**m**) neurons. n=6 (MnPO/OVLT$^{Vglut2}$), 6 (MnPO/OVLT$^{Vgat}$) mice. (**e, n**) Average of pre-ingestive responses in saline and CNO trials of mice expressing hM4Di in MnPO/OVLT$^{Vglut2}$ (**e**) and MnPO/OVLT$^{Vgat}$ (**n**) neurons. n=6 (MnPO/OVLT$^{Vglut2}$), 7 (MnPO/OVLT$^{Vgat}$) mice. (**f, o**) Data from panels (c) (**f**) and (l) (**o**) binned across drinking periods. ns, p>0.05; *p<0.05; RM two-way ANOVA, p=0.002 (MnPO/OVLT$^{Vglut2}$), p>0.05 (MnPO/OVLT$^{Vgat}$), n=6 (MnPO/OVLT$^{Vglut2}$), 7 (MnPO/OVLT$^{Vgat}$) mice. (**g, p**) Average pre- and post-ingestive responses in saline and CNO trials of mice expressing hM4Di in MnPO/OVLT$^{Vglut2}$ (**g**) and MnPO/OVLT$^{Vgat}$ (**p**) neurons. ns, p> 0.05; *p<0.05; **p<0.01; paired t-test, p=0.00474, 0.00189 (MnPO/OVLT$^{Vglut2}$), 0.0257, 0.413, 0.256 (MnPO/OVLT$^{Vgat}$), n=6 (MnPO/OVLT$^{Vglut2}$), 7 (MnPO/OVLT$^{Vgat}$) mice. (**h, q**) Effect of MnPO/OVLT$^{Vglut2}$ (**h**) and MnPO/OVLT$^{Vgat}$ (**q**) neuron inhibition on drinking and average latency to drinking onset and number of drinking bouts in saline and CNO trials. ns, p>0.05; *p<0.05; paired t-test, p=0.0179, 0.116, 0.890 (MnPO/OVLT$^{Vglut2}$), 0.6, 0.317, 0.229 (MnPO/OVLT$^{Vgat}$), n=6, 6, 5 (MnPO/OVLT$^{Vglut2}$), 5, 7, 5 (MnPO/OVLT$^{Vgat}$) mice. (**i, r**) Average change in baseline activity in saline and CNO trials of mice expressing hM4Di in MnPO/OVLT$^{Vglut2}$ (**i**) and MnPO/OVLT$^{Vgat}$ (**r**) neurons. ns, p>0.05; *p<0.05; paired t-test, p=0.0140 (MnPO/OVLT$^{Vglut2}$), 0.112 (MnPO/OVLT$^{Vgat}$), n=6 (MnPO/OVLT$^{Vglut2}$), 5 (MnPO/OVLT$^{Vgat}$) mice. Scale bar, 200 μm. Values are means ± SEMs across mice. AVP, vasopressin; MnPO, median preoptic nucleus; OVLT, organum vasculosum lamina terminalis; SFO, subfornical organ; SON, supraoptic nuclei.

The online version of this article includes the following source data for figure 5:

**Source data 1.** SON$^{AVP}$ neuron response to water bowl placement after specific inhibtion of MnPO/OVLT$^{Vglut2}$ and MnPO/OVLT$^{Vgat}$ neurons.

gradually increase as early as within 1-2 min of feeding and reaches significance at 5 min (**Mandelblat-Cerf et al., 2017**), we speculate that feeding-induced activation of SFO$^{Vglut2}$ neurons reflects systemic blood osmolality changes. Taken together, these results demonstrate that MnPO/OVLT and SFO are not involved in presystemic feeding-induced activation of SON$^{AVP}$ neurons, supporting our finding that food- and water-related presystemic regulation is mediated by non-overlapping neural circuits.

## Brainstem inputs do not mediate feeding-induced activation of AVP neurons

Given that the LT appears not to be involved in food-related presystemic regulation, we investigated other possible afferents that might provide food-related presystemic information to SON$^{AVP}$ neurons. Due to inefficiency and tropism of the rabies virus, rabies retrograde tracing can, in some cases, be prone to false-negative results (**Saleeba et al., 2019**). Therefore, we decided to study additional sites suggested by a study using a traditional retrograde tracer (**Tribollet et al., 1985**): the nucleus of the solitary tract (NTS) and A1/C1 neurons in the ventrolateral medulla (VLM). Assuming that our rabies study failed to detect these afferents, we decided to investigate whether any of these areas provide direct inputs to AVP neurons. Catecholaminergic and glutamatergic input from A1/C1 neurons to AVP neurons have been validated in previous studies (**Leng et al., 1999**; **Guyenet et al., 2013**) and therefore their connection was not tested in our study. First, we explored the projections of NTS neurons using Cre-independent ChR2 (AAV-ChR2-mCherry). Projections from the NTS were extremely sparse in the PVH and nearly absent in the SON (**Figure 7a**). Subsequent CRACM showed that the NTS do not provide direct synaptic input to AVP neurons (**Figure 7b and c**). Similar results were obtained when A2 and excitatory neurons of the NTS were studied separately (**Figure 7—figure supplement 1**), and thus the NTS was not investigated further.

To test the involvement of A1/C1 neurons in food-related presystemic regulation, we inhibited A1/C1 neurons with hM4Di while monitoring in vivo calcium dynamics of SON$^{AVP}$ neurons. To achieve specific expression of hM4Di in A1/C1 neurons, we used AVP-IRES-Cre;DBH-Flp mice in combination with AAV-fDIO-hM4Di-mCherry (**Figure 7d**). First, to test the effectiveness of hM4Di in silencing A1/C1 neurons, we studied the effect of CNO on hypotension-induced activation of AVP neurons. A1/C1 neurons are suggested to be crucial for hypotension-induced AVP release as non-specific inhibition/lesion of the VLM causes a significant attenuation of this response (**Blessing and Willoughby, 1985**; **Head et al., 1987**). A1/C1 inhibition significantly attenuated the activation of SON$^{AVP}$ neurons in response to drug-induced hypotension (**Figure 7e**). This finding validates our experimental system and provides further evidence for the importance of A1/C1 neurons in hypotension-induced AVP release. We then used the same approach to examine the involvement of A1/C1 neurons in food-related presystemic regulation. In contrast to the robust blockade of hypotension-induced SON$^{AVP}$ neuron activation, A1/C1 neuron inhibition had no effect on feeding-induced activation of SON$^{AVP}$

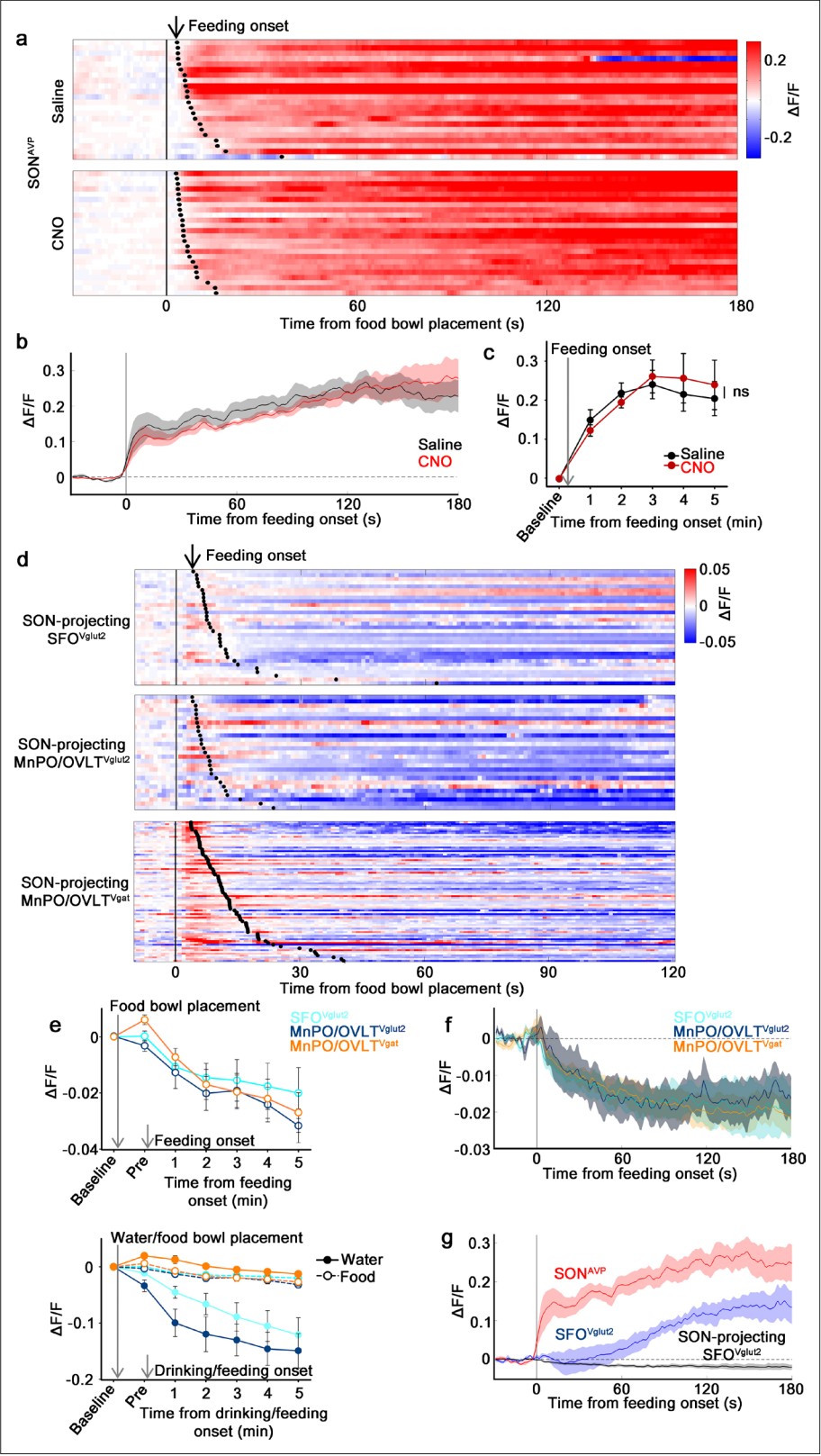

**Figure 6.** The MnPO/OVLT is not involved in food-related presystemic regulation of SON[AVP] neurons. (**a**) Single-trial timecourses of SON[AVP] population activity in response to food bowl placement in saline and CNO trials. Trials are sorted according to latency from food bowl placement to feeding onset (black ticks). n=5 mice. (**b**) Average population activity of SON[AVP] neurons in response to feeding onset in saline and CNO trials. n=5 mice. (**c**) Data

*Figure 6 continued on next page*

*Figure 6 continued*

from panel (**b**) binned across feeding periods. ns, p>0.05; RM two-way ANOVA, p>0.05, n=5 mice. (**d**) Single-trial timecourses of SON-projecting SFO$^{Vglut2}$, MnPO/OVLT$^{Vglut2}$, and MnPO/OVLT$^{Vgat}$ population activity in response to food bowl placement. Trials are sorted according to latency from food bowl placement to feeding onset (black ticks). n=7 (SFO$^{Vglut2}$), and 5 (MnPO/OVLT$^{Vglut2}$), and 12 (MnPO/OVLT$^{Vgat}$) mice. (**e**), Average population response of SON-projecting SFO$^{Vglut2}$, MnPO/OVLT$^{Vglut2}$, and MnPO/OVLT$^{Vgat}$ neurons to food bowl placement (top) and water (closed circles, solid line) versus food (open circles, dotted line) bowl placement (bottom). (**f**) Average population activity of SON-projecting SFO$^{Vglut2}$, MnPO/OVLT$^{Vglut2}$, and MnPO/OVLT$^{Vgat}$ neurons aligned to feeding onset. (**g**) Average population activity of SON$^{AVP}$, SFO$^{Vglut2}$, and SON-projecting SFO$^{Vglut2}$ neurons aligned to feeding onset. n=5 (SON$^{AVP}$, same data as **b**), 3 (SFO$^{Vglut2}$), and 7 (SON-projecting SFO$^{Vglut2}$) mice. Values are means ± SEMs across mice. AVP, vasopressin; MnPO, median preoptic nucleus; OVLT, organum vasculosum lamina terminalis; SFO, subfornical organ; SON, supraoptic nuclei.

The online version of this article includes the following source data for figure 6:

**Source data 1.** The MnPO/OVLT is not involved in food-related presystemic regulation of SON$^{AVP}$ neurons.

---

neurons (*Figure 7f*). Taken together, these data indicate that blood pressure regulation is mediated by a circuit involving A1/C1 neurons and this circuit is distinct from the food-related presystemic circuit.

## AgRP, POMC, and PNZ neurons do not mediate food-related presystemic regulation of AVP neurons

Next, we decided to test other afferents identified by rabies tracing—that is, those coming from the ARC and PNZ (*Figure 7g*, *Figure 7—figure supplement 2a*). We first decided to focus on two genetically accessible, functionally relevant neuronal populations in the ARC, AgRP and POMC neurons, which have opposite roles in regulating food intake and energy balance. To investigate whether AgRP and POMC neurons provide direct inputs to PVH$^{AVP}$ and SON$^{AVP}$ neurons, we performed projection mapping and CRACM. AgRP and POMC projections were present in the PVH and SON but no synaptic connections were detected between AgRP/POMC and AVP neurons (*Figure 7—figure supplement 3*). In line with this, specific chemogenetic inhibition of AgRP or POMC neurons had no effect on the food-related presystemic response of SON$^{AVP}$ neurons when investigated using fiber photometry in AVP-IRES-Cre;AgRP/POMC-IRES-Cre mice (*Figure 7h–j*). Short-term (<30 min following CNO injection) feeding behavior was not altered by AgRP or POMC neuron inhibition (*Figure 7—figure supplement 4* and *Krashes et al., 2011*; *Üner et al., 2019*).

Another candidate input identified by rabies mapping is the PNZ, an area surrounding the SON (*Figure 7—figure supplement 2a*). GABAergic neurons in the PNZ (PNZ$^{Vgat}$) are believed to be involved in hypertension- and hypervolemia-induced inhibition of SON$^{AVP}$ neurons (*Grindstaff and Cunningham, 2001b*; *Grindstaff and Cunningham, 2001a*; *Cunningham et al., 2002*). While we found that PNZ$^{Vgat}$ neurons provided direct inhibitory input to AVP neurons and showed a selective response to food but not water (*Figure 7—figure supplement 2b, f*), inhibition of these neurons had no effect on the baseline activity of SON$^{AVP}$ neurons (not shown) or their response to feeding (*Figure 7—figure supplement 2g,i*). Taken together, these data suggest that AgRP, POMC, and PNZ$^{Vgat}$ neurons are not primary contributors to food-related presystemic regulation.

## Food-related presystemic regulation of AVP neurons is mediated by unknown neurons in the ARC

Next, we decided to focus on the entire ARC population. Because it is difficult to selectively target all ARC neurons without hitting surrounding nuclei, to examine the role of ARC neurons we decided to use a subtractive approach by creating three groups of animals with differing hM4Di expression: (1) ARC+VMH+ DMH (*Figure 7k*), (2) VMH only (*Figure 7o*), and (3) DMH only (*Figure 7q*). This approach allowed for effective silencing of the ARC with hM4Di, which is difficult to achieve with restricted injection of Cre-independent AAVs into the ARC. All three groups of mice underwent the same experimental protocol. Mice were chronically food-restricted and were habituated to the experimental session as described above. Inhibition of neurons in all three nuclei (ARC+VMH+ DMH) caused a significant attenuation of feeding-induced activation that persisted throughout the recording (*Figure 7l–n*). In contrast, inhibition of the VMH (*Figure 7p*) or DMH (*Figure 7r*) alone, fully sparing the ARC, had no effect. Injection of CNO in all three groups did not affect the baseline activity of SON$^{AVP}$ neurons

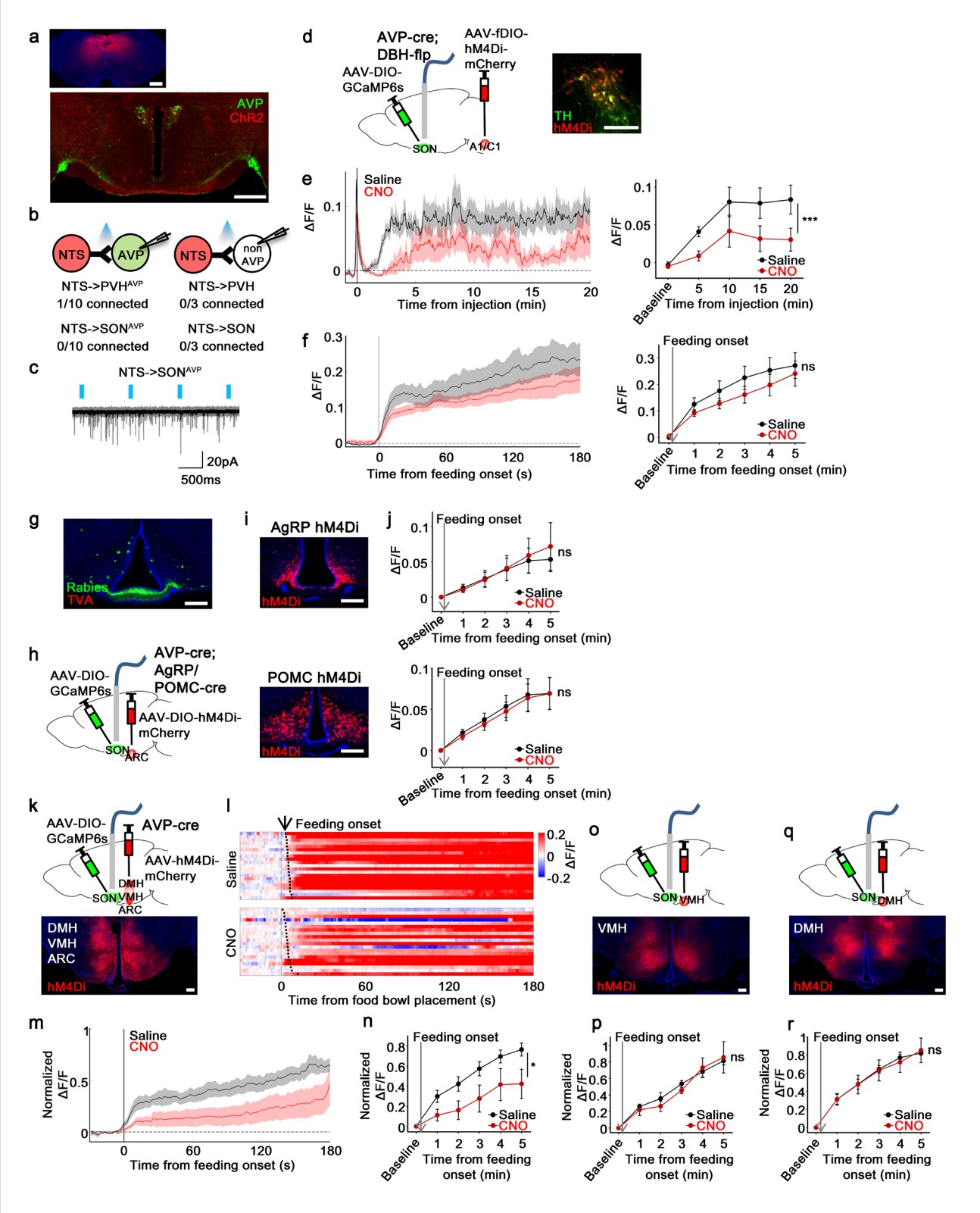

**Figure 7.** non-AgRP/POMC neurons in the ARC mediate food-related presystemic regulation of SON^AVP neurons. (**a**) Representative images showing expression of ChR2-mCherry in the NTS (top), and their efferent projections in the PVH and SON (bottom). (**b**) Number of PVH^AVP and SON^AVP neurons (left), and non-GFP PVH and SON neurons (right) receiving direct synaptic inputs from the NTS. (**c**) Representative traces showing light-evoked responses. Black trace is an average of all traces (gray) in consecutive trials. (**d**) Schematic of SON^AVP photometry experiment with hM4Di-mediated

*Figure 7 continued on next page*

*Figure 7 continued*

inhibition of A1/C1 neurons. (**e**) Average SON[AVP] population activity to hypotension induced by vasodilating drug HDZ in saline and CNO trials (left). Data binned every 5 min (right). ***p<0.001; RM two-way ANOVA, p<0.001, n=9 mice. (**f**) Average SON[AVP] population activity in response to feeding onset in saline and CNO trials (left). Data binned across feeding periods (right). ns, p>0.05; RM two-way ANOVA, p>0.05, n=16 mice. (**g**) Representative image showing rabies-labeled neurons in the ARC that are monosynaptically connected to magnocellular SON[AVP] neurons. (**h**) Schematic of SON[AVP] photometry experiment with hM4Di-mediated inhibition of AgRP or POMC neurons. (**i**) Representative images showing hM4Di expression in AgRP (top) and POMC neurons (bottom). (**j**) Average SON[AVP] population activity binned across feeding periods in saline and CNO trials of AVP-IRES-Cre;Agrp-IRES-Cre (top) and AVP-IRES-Cre;Pomc-IRES-Cre (bottom) mice. ns, p>0.05; RM two-way ANOVA, p>0.05 (AgRP), p>0.05 (POMC), n=4 (AgRP), 8 (POMC). (**k, o, q**) Schematic of SON[AVP] photometry experiment with hM4Di-mediated non-specific inhibition of the ARC+VMH+ DMH (**k**), VMH only (**o**), and DMH only (**q**). (**l**) Single-trial timecourses of SON[AVP] population activity in response to food bowl placement in saline and CNO trials of ARC+VMH+ DMH group. Trials are sorted according to latency from food bowl placement to feeding onset (black ticks). n=5 mice. (**m**) Average SON[AVP] population activity in response to feeding onset in saline and CNO trials of ARC+VMH+ DMH group. n=9 mice. (**n, p, r**) Average SON[AVP] population activity binned across feeding periods in saline and CNO trials of ARC+VMH+ DMH (**n**), VMH only (**p**), and DMH only (**r**) groups. ns, p>0.05; *p<0.05; RM two-way ANOVA, p=0.033 (ARC+VMH+ DMH), p>0.05 (VMH only), p>0.05 (DMH only), n=5 (ARC+VMH+ DMH), 4 (VMH only), 7 (DMH only) mice. Scale bars, 500 μm (**a, d**), 200 μm (**g, i, k, o, q**). Values are means ± SEMs across mice. See also *Figure 7—figure supplements 1–4*. AVP, vasopressin; DMH, dorsomedial nuclei; SON, supraoptic nuclei; VMH, ventromedial.

The online version of this article includes the following source data and figure supplement(s) for figure 7:

**Source data 1.** SON[AVP] neuron response to food bowl placement after inhibition of AgRP and POMC neurons, ARC, DMH, and VMH.

**Figure supplement 1.** The NTS does not provide input to AVP neurons.

**Figure supplement 2.** PNZ[GABA] neurons show presystemic responses to feeding but are not required for food-related presystemic regulation of SON[AVP] neurons.

**Figure supplement 2—source data 1.** PNZ[GABA] neurons show presystemic responses to feeding but are not required for food-related presystemic regulation of SON[AVP] neurons.

**Figure supplement 3.** PVH[AVP] and SON[AVP] neurons do not receive direct synaptic inputs from AgRP and POMC neurons.

**Figure supplement 4.** Inhibition of the ARC, DMH, or VMH does not affect short-term feeding behavior.

**Figure supplement 4—source data 1.** Inhibition of the ARC, DMH, or VMH does not affect short-term feeding behavior.

(not shown) or short-term feeding behavior of mice (*Figure 7—figure supplement 4*). Taken together, these results indicate that the ARC contains a cell type, distinct from AgRP or POMC neurons, that relay food-related presystemic signals to SON[AVP] neurons.

## Discussion

Water homeostasis is achieved by an intricate balance between water intake and output that are dictated, respectively, by thirst and antidiuretic hormone (AVP). While the field has experienced great advances in the neurobiological understanding of thirst in the last several years, our knowledge of neural regulation of AVP release remains rudimentary due to lack of technical approaches for reliable and temporally precise measurement of circulating AVP in vivo. In a prior study, by directly monitoring the activity of magnocellular AVP neurons, we demonstrated that AVP neuron activity, and thus likely AVP release, is altered surprisingly rapidly by water and food ingestion before any change in systemic osmolality is observed (*Mandelblat-Cerf et al., 2017*). While such feedforward, presystemic regulation has been demonstrated in other homeostatic circuits, AVP neurons are unique in that they are capable of responding presystemically to both water and food, and the response is *bidirectional*—water-predicting cues and drinking inhibit and eating activates AVP neurons—and *asymmetric*—water causes pre- and post-ingestive presystemic inhibition but food only causes post-ingestive presystemic activation. In this study, we expand on our prior findings by presenting a neural circuit mechanism by which these unique properties arise. Our study revealed two parallel, non-overlapping neural pathways for water- and food-related presystemic regulation (*Figure 8*, *Table 1*). Key components of the water-related presystemic circuit are neurons in the LT, which provide direct synaptic input to AVP neurons and exhibit water-related presystemic responses that closely resemble those of AVP neurons. Food-related presystemic regulation, on the other hand, is mediated by an as yet unidentified neuronal population in the ARC that is completely distinct from feeding-regulating AgRP and POMC neurons. In the course of our study, we confirmed additional afferents that were previously proposed to be involved in other aspects of AVP function, such as A1/C1 neurons in the brainstem that mediate hypotension-induced AVP release (*Blessing and Willoughby, 1985*; *Head et al., 1987*; *Leng et al.,*

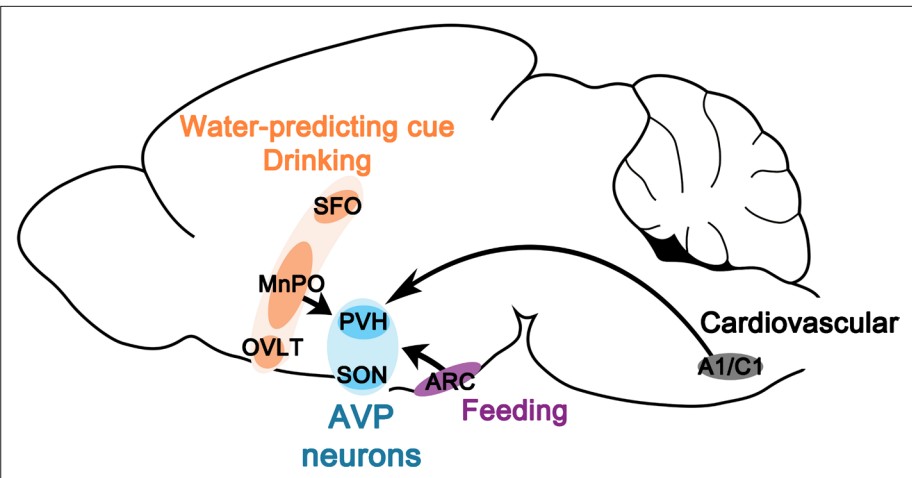

**Figure 8.** Neural circuits for presystemic regulation of AVP neurons. AVP, vasopressin; MnPO, median preoptic nucleus; OVLT, organum vasculosum lamina terminalis; PVH, paraventricular; SFO, subfornical organ SON, supraoptic nuclei.

*1999*; *Guyenet et al., 2013*). Collectively, these findings demonstrate that water- versus food-related presystemic signals are relayed to AVP neurons independently and differently via two non-overlapping circuits to give rise to the bidirectional, asymmetric presystemic response of AVP neurons.

Recent studies have identified multiple molecularly or anatomically distinct neuronal populations in the LT that are involved in different aspects of water homeostasis ( *Abbott et al., 2016*; *Allen et al., 2017*; *Matsuda et al., 2017*; *Augustine et al., 2018*; *Matsuda et al., 2020*; *Pool et al., 2020*). In this study, we investigated the role of the LT neurons in mediating presystemic regulation of AVP neurons. We identified at least three groups of LT neurons that provide direct input to AVP neurons: SFO$^{Vglut2}$, MnPO/OVLT$^{Vglut2}$, and MnPO/OVLT$^{Vgat}$ neurons. Interestingly, while all these neurons connect to AVP neurons with high probability, only MnPO/OVLT neurons were required for their presystemic regulation. More strikingly, while being indispensable for water-related presystemic responses of AVP neurons, MnPO/OVLT neurons played no role in mediating food-related presystemic responses, indicating that presystemic regulation of AVP neurons involves two distinct neural pathways dedicated to water- or food-related regulation. Three groups of AVP-regulating LT neurons were studied individually by selectively isolating them using projection-specific approaches (i.e., to the SON) and genetic markers (i.e., Vglut2 and Vgat). As can be expected from their opposite neurochemical nature, SON-projecting MnPO/OVLT$^{Vglut2}$ and MnPO/OVLT$^{Vgat}$ neurons showed opposing responses to water-related presystemic signals. SON-projecting MnPO/OVLT$^{Vglut2}$ neurons were rapidly suppressed by both water-predicting cues and drinking whereas SON-projecting MnPO/OVLT$^{Vgat}$ neurons were activated transiently and exclusively by water-predicting cues. SON-projecting SFO$^{Vglut2}$ neurons showed similar presystemic response to MnPO/OVLT$^{Vglut2}$ neurons but their contribution to presystemic regulation of AVP neurons may be minor as inhibition of the SFO did not affect the activity of AVP neurons. Of note, while we refer to SON-projecting LT neurons as 'AVP-regulating' based on their high probability connections to SON$^{AVP}$ neurons, we acknowledge that subsets of SON-projecting LT neurons may also be involved in regulation of other downstream processes such as oxytocin release (via their connection to SON$^{Oxytocin}$ (non-AVP) neurons as demonstrated by CRACM) and thirst (via their collaterals to other downstream structures as demonstrated by collateral mapping).

By selectively silencing MnPO/OVLT$^{Vglut2}$ and MnPO/OVLT$^{Vgat}$ neurons, we demonstrate that both excitatory and inhibitory inputs from the MnPO/OVLT are necessary for water-related presystemic regulation of AVP neurons. MnPO/OVLT$^{Vgat}$ and MnPO/OVLT$^{Vglut2}$ neurons were each responsible for pre- and post-ingestive phases of the response, respectively, which is consistent with the contrasting neural dynamics of SON-projecting MnPO/OVLT$^{Vgat}$ and MnPO/OVLT$^{Vglut2}$ neurons. Initial pre-ingestive cue-induced suppression of AVP neurons was mediated primarily by MnPO/OVLT$^{Vgat}$ neurons as their silencing completely prevented this response. Silencing of MnPO/OVLT$^{Vglut2}$ neurons also caused a significant attenuation but failed to completely block the pre-ingestive response. These results suggest that the pre-ingestive response is a combined outcome of direct and indirect suppression

**Table 1.** Summary of the result.

| Afferents tested | Connection to AVP neurons (validation method)* | Presystemic response (pre-/post-ingestive) | | Effect of inhibition on AVP neuron activity |
|---|---|---|---|---|
| | | Water | Food | |
| SON-projecting SFO$^{Vglut2}$, MnPO/OVLT$^{Vglut2}$ | ✓ (Rabies mapping, CRACM, and projection mapping) | ↓/↓↓ | ↔/↔ | ▼ Water-related presystemic response (pre- and post-ingestive) |
| SON-projecting MnPO/OVLT$^{Vgat}$ | ✓ (Rabies mapping, CRACM, and projection mapping) | ↑/↔ | ↑/↔ | ▼ Water-related presystemic response (pre-ingestive only) |
| A1/C1 | ✓ (CRACM and projection mapping) | ? | ? | ▼ Hypotension-induced activation |
| PNZ$^{Vgat}$ | ✓ (Rabies mapping and CRACM) | ↑/↔ | ↑/↓↓ | No effect |
| AgRP$^{a}$ | X (Rabies mapping, CRACM, and projection mapping) | ↔ | ↓/↓↓ | No effect |
| POMC$^{a}$ | X (Rabies mapping, CRACM, and projection mapping) | ? | ↑/↑↑ | No effect |
| ARC | ✓ (Rabies mapping) | ? | ? | ▼ Feeding-related presystemic response |

*, connected; X, not connected; ↑, increase (↑<↑↑); ↓, decrease (↓<↓↓); ↔, no change;?, not tested; ▼, significantly attenuated; a, *Betley et al., 2015*; *Chen et al., 2015*; *Mandelblat-Cerf et al., 2017*.

of AVP neurons by MnPO/OVLT$^{Vgat}$ neurons, via their direct projections to AVP neurons and local projections to MnPO/OVLT$^{Vglut2}$ neurons, which consequently results in reduced excitatory drive to AVP neurons. Post-ingestive drinking-induced suppression of AVP neurons was primarily the result of decreased excitatory drive from the MnPO/OVLT as post-ingestive response was significantly attenuated by silencing of MnPO/OVLT$^{Vglut2}$ neurons, but not MnPO/OVLT$^{Vgat}$ neurons. Silencing of MnPO/OVLT$^{Vglut2}$ neurons simultaneously caused a significant reduction in the amount of water intake. While reduced water intake could contribute to the reduction in post-ingestive response, this is unlikely a major factor given the consistent degree of silencing throughout both pre- and post-ingestive phases and the high probability connection and similar response dynamics in SON-projecting MnPO/OVLT$^{Vglut2}$ versus SON$^{AVP}$ neurons.

Presystemic responses of thirst-regulating counterparts of the LT have been extensively studied by many groups (*Zimmerman et al., 2016*; *Allen et al., 2017*; *Augustine et al., 2018*; *Augustine et al., 2019*; *Zimmerman et al., 2019*). While thirst-regulating LT neurons also exhibit presystemic suppression (thirst-promoting neurons) and activation (thirst-suppressing neurons), their responses surprisingly lack a pre-ingestive component that is present in AVP-regulating LT neurons and also hunger-regulating AgRP and POMC neurons (this study and *Betley et al., 2015*; *Chen et al., 2015*; *Mandelblat-Cerf et al., 2017*). This raises the possibility that presystemic regulation of thirst neurons may be different. Alternatively, it is possible that prior studies missed this regulation due to either its magnitude being small in comparison to post-ingestive regulation and/or the fact that population recordings, which may have pooled thirst-regulating LT neurons with other neurons, obscured its detection. Indeed, a study using single-cell imaging reported a small subset of MnPO/OVLT neurons that are activated or inhibited by pre-ingestive cue (*Zimmerman et al., 2019*). In addition, considering that the pre-ingestive response is not innate but requires several days of training (*Figure 2— figure supplement 1* and *Mandelblat-Cerf et al., 2017*), failure to detect pre-ingestive responses in these thirst-related studies might possibly be due to the use of experimental paradigms not optimally designed to assess pre-ingestive responses.

The MnPO is traditionally considered as a hub that functions as the main input and output region of the LT (*McKinley et al., 2015*). In line with this, we found that SON-projecting MnPO/OVLT $^{Vglut2}$ neurons receive inputs from multiple regions outside the LT while inputs to SON-projecting SFO$^{Vglut2}$ neurons were confined to the MnPO/OVLT. This result supports hierarchical organization of the LT with the MnPO/OVLT functioning as the main entry point for neurally transmitted afferent information. Systemic osmolality information (from the CVOs [the SFO and OVLT]) and water-related presystemic information (from yet-undefined regions outside the LT) converge on the MnPO/OVLT and are redistributed to AVP neurons and other downstream targets directly, or indirectly via the SFO. Based on our rabies mapping of afferents to SON-projecting MnPO/OVLT$^{Vglut2}$ neurons, the following sites could in principle relay water-related presystemic information to the MnPO/OVLT: the VMH, DMH, PVH, and LPBN. The DMH is of particular interest as it is already known to play a role in presystemic regulation of AgRP neurons (*Garfield et al., 2016*). By analogy, it may also contain neurons involved in water-related presystemic regulation of MnPO/OVLT neurons. The LPBN is also of interest in that it is known to relay interoceptive information to forebrain sites (*Herbert et al., 1990*), and it has been strongly implicated in regulation of water balance (*Davern, 2014*; *Gizowski and Bourque, 2018*). Indeed, recent studies identified two groups of thirst-suppressing neurons in the PBN that send direct projections to the MnPO/OVLT and exhibit post-ingestive activation in response to drinking (*Ryan et al., 2017*; *Kim et al., 2020*).

We found that food-related presystemic regulation is not mediated by the LT. LT neurons do not show food-related presystemic changes in their activity and silencing of LT neurons does not affect food-related presystemic changes in activity of AVP neurons. Inhibition of neurons in the ARC, on the other hand, causes a significant attenuation of food-related presystemic responses suggesting that neurons in this region mediate food-related regulation. Many anatomical and electrophysiological studies have demonstrated that AVP neurons receive direct excitatory and inhibitory projections from the ARC (*Iijima and Ogawa, 1981*; *Saphier and Feldman, 1986*; *Leng et al., 1988*; *Ludwig and Leng, 2000*; *Pineda et al., 2016*). This is further supported by our rabies mapping results. Due to technical limitations, however, the ARC neurons responsible for food-related presystemic regulation were not identified. The most widely studied neurons in the ARC, AgRP and POMC neurons, were ruled out, as inhibition of these neurons had no effect on the food-related presystemic response. Considering

rapid feeding-induced activation of AVP neurons, it is highly likely that feeding-related presystemic regulation involves glutamatergic neurons in the ARC. Oxytocin receptor-expressing glutamatergic neurons (*Fenselau et al., 2017*) are a good candidate for mediating food-related presystemic regulation as these neurons are activated by feeding and are capable of inhibiting food intake rapidly when activated. Alternatively, it is possible that a completely novel ARC neuron is involved (*Campbell et al., 2017*). The combination of projection- and synaptic connectivity-based gene profiling techniques will facilitate specific identification of ARC neurons mediating food-related presystemic regulation.

In summary, presystemic regulation of AVP release is mediated by the LT and ARC circuits that, respectively, exclusively relay water- or food-related presystemic signals. The two circuits operate largely independently and relay different repertoire of presystemic signals (water-related pre- and post-ingestive signals by the LT and food-related post-ingestive signals by the ARC), giving rise to the bidirectional and asymmetric nature of AVP neurons' presystemic responses. Additional anatomically and functionally distinct neural circuits converge on AVP neurons to mediate other aspects of AVP function, such as occurs during hypotension. Convergence and integration of multiple inputs at the level of AVP neurons, the final common pathway for AVP release, provide a neural circuit basis for differential regulation of AVP release by diverse behavioral and physiological stimuli. Identifying and characterizing key neural components of the extended neural circuitry underlying these processes will be an important area for future investigation.

# Materials and methods

**Key resources table**

| Reagent type (species) or resource | Designation | Source or reference | Identifiers | Additional information |
|---|---|---|---|---|
| Strain, strain background (*Mus musculus*) | *Avp*-IRES-Cre | PMID:24634830 | | |
| Strain, strain background (*M. musculus*) | *Slc17a6*-IRES-cre | The Jackson Laboratory | RRID:IMSR_JAX:016963 | PMID:21745644 |
| Strain, strain background (*M. musculus*) | *Slc32a1*-IRES-Cre | The Jackson Laboratory | RRID:IMSR_JAX:016962 | PMID:21745644 |
| Strain, strain background (*M. musculus*) | *Nos1*-IRES-cre | The Jackson Laboratory | RRID:IMSR_JAX:017526 | PMID:22522563 |
| Strain, strain background (*M. musculus*) | *Slc17a6*-IRES-Flp | The Jackson Laboratory | RRID:IMSR_JAX:030212 | |
| Strain, strain background (*M. musculus*) | *Slc32a1*-IRES-Flp | The Jackson Laboratory | RRID:IMSR_JAX:031331 | |
| Strain, strain background (*M. musculus*) | *Th*-Cre | The Jackson Laboratory | RRID:IMSR_JAX:008601 | PMID:16033881 |
| Strain, strain background (*M. musculus*) | *Dbh*-Flp | MMRRC | RRID:MMRRC_041575 | |
| Strain, strain background (*M. musculus*) | *Avp*-GFP | MMRRC | RRID:MMRRC_015858-UCD | |
| Strain, strain background (*M. musculus*) | *Agrp*-IRES-Cre | The Jackson Laboratory | RRID:IMSR_JAX:012899 | |
| Strain, strain background (*M. musculus*) | *Pomc*-IRES-Cre | PMID:27869800 | | |
| Genetic reagent (*Adeno-associated virus*) | AAV8-FLEX-TVA-mCherry | UNC Vector Core | RRID:Addgene_38044 | |
| Genetic reagent (*Adeno-associated virus*) | AAV1-CAG-FLEX-RG | UNC Vector Core | RRID:Addgene_48333 | |
| Genetic reagent (*Adeno-associated virus*) | SADΔG–EGFP (EnvA) rabies | Salk Gene Transfer Targeting and Therapeutics Core | RRID:Addgene_32635 | |

*Continued on next page*

*Continued*

| Reagent type (species) or resource | Designation | Source or reference | Identifiers | Additional information |
|---|---|---|---|---|
| Genetic reagent (*Adeno-associated virus*) | AAV9-EF1$\alpha$-DIO-ChR2(H134R)-mCherry | Penn Vector Core | RRID:Addgene_20297 | |
| Genetic reagent (*Adeno-associated virus*) | AAV8.2-hEF1a-synaptophysin-EYFP | MGH Gene Delivery Technology Core | RN9 | |
| Genetic reagent (*Adeno-associated virus*) | AAV9-CAG-ChR2(H134R)-mCherry | Penn Vector Core | RRID:Addgene_100054 | |
| Genetic reagent (*Adeno-associated virus*) | HSV-hEF1a-LS1L-GCaMP6s | MGH Gene Delivery Technology Core | RN507 | |
| Genetic reagent (*Adeno-associated virus*) | AAV1-Syn-FLEX-GCaMP6s | Penn Vector Core | RRID:Addgene_100845 | |
| Genetic reagent (*Adeno-associated virus*) | AAV1-Ef1a-DIO-EYFP | UNC Vector Core | RRID:Addgene_27056 | |
| Genetic reagent (*Adeno-associated virus*) | AAV8-hSyn-hM4Di-mCherry | UNC Vector Core | RRID:Addgene_44362 | |
| Genetic reagent (*Adeno-associated virus*) | AAV8-nEF-fDIO-hM4Di-mCherry | This paper | N/A | |
| Antibody | Anti-mCherry (rat monoclonal) | Life Technologies | M11217 | IF (1:1000) |
| Antibody | Anti-dsRed (rabbit polyclonal) | Clontech | 632,496 | IF (1:1000) |
| Antibody | Anti-GFP (chicken polyclonal) | Life Technologies | A10262 | IF (1:1000) |
| Antibody | Anti-vasopressin (rabbit polyclonal) | Sigma-Aldrich | AB1565 | IF (1:1000) |
| Antibody | Anti-TH (rabbit polyclonal) | Millipore | AB152 | IF (1:1000) |
| Antibody | Anti-POMC precursor (rabbit polyclonal) | Phoenix Pharmaceuticals | H-029–30 | IF (1:1000) |
| Antibody | Anti-AgRP (goat polyclonal) | Neuromics | GT15023 | IF (1:1000) |
| Software, algorithm | SigmaPlot | Systat Software Inc | RRID:SCR_003210 | |
| Software, algorithm | MATLAB | MathWorks | RRID:SCR_001622 | |

## Mice

Animals were housed at 22–24°C on a 12:12 light/dark cycle (light cycle: 6:00 a.m. to 6:00 p.m.) with standard mouse chow (Teklad F6 Rodent Diet 8664; Harlan Teklad) and water provided ad libitum, unless specified otherwise. Adult male and female mice (8–16 weeks old) were used for experiments. Mice were maintained on a mixed background. Transgenic mouse strains used: *Avp*-IRES-Cre (*Pei et al., 2014*), *Slc17a6*-IRES-Cre (Vglut2-IRES-Cre) (*Vong et al., 2011*) (Jackson Labs Stock 016963), *Slc32a1*-IRES-Cre (Vgat-IRES-Cre) (*Vong et al., 2011*) (Jackson Labs stock 016962), *Nos1*-IRES-Cre (Jackson Labs Stock 017526), *Slc17a6*-IRES-Flp (Vglut2-IRES2-FLPo-D) (Jackson Labs Stock 030212), *Slc32a1*-IRES-Flp (Vgat-IRES2-FLPo-D) (Jackson Labs Stock 031331), *Th*-Cre (*Savitt et al., 2005*) (Jackson Labs Stock 008601), *Dbh*-Flp (MMRRC 041575), *Avp*-GFP (MMRRC 015858-UCD), *Agrp*-IRES-Cre (*Tong et al., 2008*) (Jackson Labs Stock 012899), and *Pomc*-IRES-Cre (*Fenselau et al., 2017*).

## Recombinant adeno-associated viral vectors

The following viral vectors were used in this study: AAV8-FLEX-TVA-mCherry (UNC Vector Core, Addgene 38044), AAV1-CAG-FLEX-RG (UNC Vector Core, Addgene 48333), SADΔG–EGFP (EnvA) rabies (Salk Gene Transfer Targeting and Therapeutics Core, Addgene 32635), AAV9-EF1α-DIO-ChR2(H134R)-mCherry (Penn Vector Core, Addgene 20297), AAV8-EF1a-DIO-synaptophysin-YFP (MIT Vector Core), AAV9-CAG-ChR2(H134R)-mCherry (Penn Vector Core, Addgene 100054), HSV-hEF1a-LS1L-GCaMP6s (MGH Gene Delivery Technology Core, RN507), AAV1-Syn-FLEX-GCaMP6s (Penn Vector Core, Addgene 100845), AAV1-Ef1a-DIO-EYFP (UNC Vector Core, Addgene 27056), AAV8-hSyn-DIO-hM4Di-mCherry (UNC Vector Core, Addgene 44362), and AAV8-nEF-fDIO-hM4Di-mCherry

(Boston Children's Hospital Viral Core, Custom-made vector [hM4Di-mCherry was cut from AAV8-hSyn-DIO-hM4Di-mCherry {Addgene 44362} by AscI and NheI, and subcloned into pAAV-nEF Con/Fon hChR2(H134R)-EYFP {Addgene 55644} into AscI and XbaI]).

## Stereotaxic surgery and viral injections

For viral injections, mice were anaesthetized with ketamine/xylazine (100 and 10 mg/kg, respectively, i.p.) and then placed in a stereotaxic apparatus (David Kopf model 940). A pulled glass micropipette (20–40 μm diameter tip) was used for stereotaxic injections of adeno-associated virus (AAV). Virus was injected into the posterior pituitary (100 nl; AP: –3.0 mm; ML: ± 0 mm; DV: –6.0 mm from bregma), SON/PNZ (100 nl/side; AP: –0.65 mm; ML: ± 1.25 mm; DV: –5.4 mm from bregma), MnPO/OVLT (10 nl/depth; AP: +0.25 mm; ML: 0 mm; DV: –4.8, 4.5 mm from bregma), SFO (10 nl; AP: –0.6 mm; ML: 0 mm; DV: –2.6 mm from bregma), PVH (50 nl/side; AP: –0.75 mm; ML: ± 0.3 mm; DV: –4.85 mm from bregma), DMH (50 nl/side; AP: –1.85 mm; ML: ± 0.3 mm; DV: –5.2 mm from bregma), VMH (50 nl/side; AP: –1.6 mm; ML: ± 0.4 mm; DV: –5.5 mm from bregma), or Arcuate (50 nl/side; AP: –1.5 mm; ML: ± 0.25 mm; DV: –5.8 mm from bregma) by an air pressure system using picoliter air puffs through a solenoid valve (Clippard EV 24VDC) pulsed by a Grass S48 stimulator to control injection speed (40 nl/min). The pipette was removed 3 min post-injection followed by wound closure using tissue adhesive (3 M Vetbond). For viral injections into the NTS and VLM, mice were placed into a stereotaxic apparatus with the head angled down at approximately 45°. An incision was made at the level of the cisterna magna, then skin and muscle were retracted to expose the dura mater covering the fourth ventricle. A 28-gauge needle was used to make an incision in the dura and allow access to the NTS and VLM. Virus was then injected into the NTS (10 nl/side; AP: –0.2 mm; ML: ± 0.2 mm; DV: –0.2 mm from obex) VLM (50 nl*2/side; AP: –0.3 and –0.6 mm; ML: ± 1.3 mm; DV: –1.7 mm from obex) as described above. The pipette was removed 3 min post-injection followed by wound closure using absorbable suture for muscle and silk suture for skin. For fiber photometry, an optic fiber (200 μm diameter, NA=0.39, metal ferrule, Thorlabs) was implanted in the MnPO/OVLT, SFO, or SON and secured to the skull with dental cement. Subcutaneous injection of sustained release Meloxicam (4 mg/kg) was provided as postoperative care. The mouse was kept in a warm environment and closely monitored until resuming normal activity.

## Brain slice electrophysiology and channelrhodopsin-assisted circuit mapping

To prepare brain slices for electrophysiological recordings, brains were removed from anesthetized mice and immediately placed in ice-cold cutting solution consisting of (in mM): 72 sucrose, 83 NaCl, 2.5 KCl, 1 NaH2PO4, 26 NaHCO3, 22 glucose, 5 MgCl2, 1 CaCl2, oxygenated with 95 % O2/5 % CO2, measured osmolarity 310–320 mOsm/l. Cutting solution was prepared and used within 72 hr. About 250-μm-thick coronal sections containing the PVH and SON were cut with a vibratome (7000smz-2 Campden Instruments) and incubated in an oxygenated cutting solution at 34 °C for 25 min. The SON was located by using the bifurcation of the anterior and middle cerebral arteries on the ventral surface of the brain as a landmark. Slices were transferred to oxygenated aCSF (126 mM NaCl, 21.4 mM NaHCO3, 2.5 mM KCl, 1.2 mM NaH2PO4, 1.2 mM MgCl2, 2.4 mM CaCl2, and 10 mM glucose) and stored in the same solution at room temperature (20–24°C) for at least 60 min prior to recording. A single slice was placed in the recording chamber where it was continuously super-fused at a rate of 3–4 ml per min with oxygenated aCSF. Neurons were visualized with an upright microscope equipped with infrared-differential interference contrast and fluorescence optics. Borosilicate glass microelectrodes (5–7 MΩ) were filled with internal solution. Whole-cell voltage clamp recordings were obtained using Cs-based internal solutions containing either (in mM): 135 CsMeSO3, 10 HEPES, 1 EGTA, 4 MgCl2, 4 Na2-ATP, 0.4 Na2-GTP, and 10 Na2-phosphocreatine (pH 7.3; 295 mOsm) for recording EPSCs; or for recording IPSCs from Vgat neurons expressing ChR2: 140 CsCl, 1 BAPTA, 10 HEPES, 5 MgCl2, 5 Mg-ATP, 0.3 Na2GTP, and 10 lidocaine N-ethyl bromide ([QX-314], pH 7.35, and 290 mOsm). To photostimulate ChR2-positive fibers, an LED light source (473 nm) was used. The blue light was focused on to the back aperture of the microscope objective, producing a wide-field exposure around the recorded cell of 1 mW. The light power at the specimen was measured using an optical power meter PM100D (Thorlabs). The light output is controlled by a programmable pulse stimulator, Master-8 (AMPI Co., Israel), and the pClamp 10.2 software (AXON Instruments). All recordings were

made using Multiclamp 700B amplifier, and data was filtered at 2 kHz and digitized at 10 kHz. The photostimulation-evoked EPSC detection protocol consisted of four blue light laser pulses administered 1 s apart during the first 4 s of an 8 s sweep, repeated for a total of 30 sweeps. We attempted to maximize our ability to detect light-evoked currents by biasing our recordings to cell bodies within the densest axon fields. In some experiments, TTX (1 mM) and 4-AP (100 mM) were added to the bath solution to confirm monosynaptic connectivity. All CRACM results presented are from 2 to 3 mice per group. All analysis was conducted off-line in Clampfit 10 and Origin.

## Fiber photometry experiments and analysis of photometry data

All experiments were conducted in the home-cage in freely moving mice. Beginning 3 weeks post-surgery (details above), animals prepared for in vivo fiber photometry experiments (outlined above), were food restricted to 85–90% of initial body weight. Over this 1-week period, mice were acclimated to chow pellets (500 mg, Bio-Serv Dustless Precision Pellets), and to the food bowl and water bowl used in subsequent photometry experiments. Mice were habituated to the paradigm for at least 5 days prior to the first recording day. For experiments with hM4Di, saline or CNO (1 mg/kg) was injected 20 min prior to the experiment. In vivo fiber photometry was conducted as previously described (*Mandelblat-Cerf et al., 2017*). A fiber optic cable (1 m long, metal ferrule, 400 µm diameter; Doric Lenses) was attached to the implanted optic cannula with zirconia sleeves (Doric Lenses). Laser light (473 nm) was focused on the opposite end of the fiber optic cable to titrate the light intensity entering the brain to 0.1–0.2 mW. Emitted light was passed through a dichroic mirror (Di02-R488-25×36, Semrock) and GFP emission filter (FF03-525/50-25, Semrock), before being focused onto a sensitive photodetector (Newport part #2151). The GCaMP6 signal was passed through a low-pass filter (50 Hz), and digitized at 1 kHz using a National Instruments data acquisition card and MATLAB software.

The recorded data was exported and then imported into MATLAB for analysis. Fluorescent traces were down-sampled to 1 Hz. We calculated the fractional change in GCaMP6s fluorescence according to the following equation: $\Delta F/F=(F-F0)/F0$, where F is fluorescence measurement and F0 is the mean fluorescence in the 30 s prior to food/water bowl presentation or feeding onset or 2 or 3 min prior to drug injection. In some experiments, to facilitate the comparison across different groups, we normalized the responses of each mouse. In *Figure 4f*, normalization was performed by $(\Delta F/F)/F_{total\Delta F}$, where $F_{total\Delta F}$ was maximum average change over the entire recording period. In *Figures 5 and 7*, normalization was performed by $(\Delta F/F)/c$, where c was maximum average response of control trials within 5 min of water bowl placement or feeding onset.

## Drinking behavior

Drinking was assessed by modified Drinking Event Monitor cages (Lickometer cages; Columbus Instruments). Mice were placed into these "lickometer" cages where licks on each bottle sipper tube were detected by electrical conductivity and recorded by a computer counter interface. CNO or Saline was injected 15 min prior to the onset of the experiment.

## Brain tissue preparation

Animals were terminally anesthetized with 7 % chloral hydrate diluted in saline (350 mg/kg) and transcardially perfused with phosphate-buffered saline (PBS) followed by 10 % neutral buffered formalin (PFA). Brains were removed, stored in the same fixative overnight, transferred into 20 % sucrose at 4 °C overnight, and cut into 40 µm sections on a freezing microtome (Leica) coronally into two equal series.

## Immunohistochemistry

Brain sections were washed in PBS with Tween-20, pH 7.4 (PBST), and blocked in 3 % normal donkey serum in PBST for 1 h at room temperature. Brain sections were then incubated overnight at room temperature in blocking solution containing primary antiserum (rat anti-mCherry, Life Technologies M11217, 1:1000; rabbit anti-dsRed, Clontech 632496, 1:1000; chicken anti-GFP, Life Technologies A10262, 1:1000; rabbit anti-vasopressin, Sigma-Aldrich AB1565, 1:1000; rabbit anti-TH, Millipore AB152, 1:1000; rabbit anti-POMC precursor, Phoenix Pharmaceuticals H-029–30, 1:1000; goat anti-AgRP, Neuromics GT15023, 1:1,000). The next morning sections were extensively washed in PBS and

then incubated in Alexa-fluorophore secondary antibody (1:1000) for 1 hr at room temperature. After several washes in PBS, sections were mounted on gelatin-coated slides and fluorescence images were captured with Olympus VS120 slide scanner microscope.

## Statistical analysis

Statistical analyses were performed using SigmaPlot software. Electrophysiological traces were analyzed on Clamp fit 10 (Molecular Devices) and Origin (Origin Lab) software. No statistical method was used to predetermine sample size. Blinding methods were not used. All data presented met the assumptions of the statistical test employed. Specific statistical tests are specified in the figure legends. Animals were excluded from analysis if histological validation revealed poor or inaccurate reporter expression or inaccurate fiber placement unless otherwise noted. n values reflect the final number of validated animals per group included in the analysis.

## Acknowledgements

The authors would like to thank Drs. C Saper, J Majzoub, S Liberles for helpful discussion; Y Li and Z Yang for technical assistance; J Lawitts at the BIDMC Transgenic Core; C Wang and Y Zhang at BCH Viral Core. This research was funded by NIH F31 DK109575 (AK), NIH R01 DK075632, R01 DK096010, R01 DK089044, R01 DK111401, P30 DK046200, and P30 DK057521 (BBL), NIH New Innovator Award DP2 DK105570, R01 DK109930, McKnight Scholar Award, and Pew Scholar Award (MLA).

## Additional information

### Funding

| Funder | Grant reference number | Author |
| --- | --- | --- |
| National Institute of Diabetes and Digestive and Kidney Diseases | F31 DK109575 | Angela Kim |
| National Institute of Diabetes and Digestive and Kidney Diseases | R01 DK075632 | Bradford B Lowell |
| National Institute of Diabetes and Digestive and Kidney Diseases | R01 DK096010 | Bradford B Lowell |
| National Institute of Diabetes and Digestive and Kidney Diseases | R01 DK089044 | Bradford B Lowell |
| National Institute of Diabetes and Digestive and Kidney Diseases | R01 DK111401 | Bradford B Lowell |
| National Institute of Diabetes and Digestive and Kidney Diseases | P30 DK046200 | Bradford B Lowell |
| National Institute of Diabetes and Digestive and Kidney Diseases | DP2 DK105570 | Mark L Andermann |
| National Institute of Diabetes and Digestive and Kidney Diseases | R01 DK109930 | Mark L Andermann |
| McKnight Foundation | McKnight Scholar Award | Mark L Andermann |
| Pew Charitable Trusts | Pew Scholar Award | Mark L Andermann |
| National Institute of Diabetes and Digestive and Kidney Diseases | P30 DK057521 | Bradford B Lowell |

| Funder | Grant reference number | Author |
|---|---|---|

The funders had no role in study design, data collection and interpretation, or the decision to submit the work for publication.

## Author contributions

Angela Kim, Conceptualization, Data curation, Formal analysis, Funding acquisition, Investigation, Methodology, Project administration, Resources, Software, Supervision, Validation, Visualization, Writing - original draft, Writing - review and editing; Joseph C Madara, Data curation, Formal analysis, Investigation, Writing - review and editing; Chen Wu, Investigation, Methodology, Resources, Writing - review and editing; Mark L Andermann, Conceptualization, Data curation, Resources, Supervision, Writing - review and editing; Bradford B Lowell, Conceptualization, Data curation, Funding acquisition, Project administration, Resources, Supervision, Writing - original draft, Writing - review and editing

## Author ORCIDs

Angela Kim  http://orcid.org/0000-0002-9475-0798
Mark L Andermann  http://orcid.org/0000-0002-9882-933X
Bradford B Lowell  http://orcid.org/0000-0002-0436-3760

## Ethics

All animal care and experimental procedures were approved in advance by the National Institute of Health and Beth Israel Deaconess Medical Center Institutional Animal Care and Use Committee.

## Decision letter and Author response

Decision letter https://doi.org/10.7554/66609.sa1
Author response https://doi.org/10.7554/66609.sa2

## Additional files

### Supplementary files

• Transparent reporting form

### Data availability

All data generated or analysed during this study are included in the manuscript and supporting files.

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
