## [Decision Letter]

**Acceptance summary:**

This manuscript reveals elements of the neural circuitry controlling the activity of vasopressin-expressing (VP) endocrine output neurons in the thirst circuit that enables them to anticipate systemic osmotic challenges triggered by drinking and food intake. Importantly, the authors show that the drinking and feeding related effects on VP neural activity originate from distinct anatomical sites thus laying the groundwork for mapping out the rest of the neural circuit responsible for anticipatory regulation of thirst.

**Decision letter after peer review:**

Thank you for submitting your article "Neural Basis for Regulation of Vasopressin Secretion by Anticipated Disturbances in Osmolality" for consideration by *eLife*. Your article has been reviewed by 3 peer reviewers, including Richard D Palmiter as Reviewing Editor and Reviewer #1, and the evaluation has been overseen by Catherine Dulac as the Senior Editor. The following individuals involved in review of your submission have agreed to reveal their identity: Zachary Knight, PhD (Reviewer #2); Yuki Oka (Reviewer #3).

Essential revisions:

1. It is not clearly reported to what extent the chemogenetic silencing of the MnPO/OVLT in the mice used in Figure 2 and 5 reduces the amount of water consumed and how this relates to the dynamics in each animal/trial. This is confounding in two ways. Given that the silencing does not fully block drinking, this implies that the MnPO/OVLT silencing is incomplete (based on Augustine 2018), and thus the negative photometry result in Figure 5 is hard to interpret. Conversely, if silencing reduces drinking partially, which seems likely, then this behavioral change could account for the reduced presystemic inhibition of AVP neurons. It is hard to see how the direct effect of MnPO on AVP neural dynamics could be separated from its effects on behavior in this experiment.

2. MnPO neurons are heterogeneous in their dynamics, especially the MnPO-GABA neurons, and for this reason ruling out a possible mechanism based on a photometry trace is challenging. For example, compare the interpretation of the photometry recordings of MnPO-Glp1r neurons in (Augustine, 2018) with the results of single cell imaging of the same neurons in (Zimmerman, 2019). -In Figure 2, DREADD was used to suppress the activity of the LT. However, the virus construct is a general promoter, and no data are provided to demonstrate that CNO/DREADD works in this system or cells. Because there are no behavioral effects of CNO inhibition of SFO or MnPO/OVLT, the authors should confirm the efficiency of the chemogenetic manipulation.

3. Although the authors revealed the anatomic sites relevant for different kinds of presystemic regulation of VP neurons, the causal role of specific cell types in these structures that provide this input remains untested/unclear. The manuscript would be significantly more impactful if they tested whether excitatory and inhibitory populations in the MnPO/OVLT indeed mediate the pre- and post-ingestive effects on pre-systemic VP neuronal activity. If these data are in hand or can be generated in a timely manner, they should be included. Otherwise, discuss the caveats and tone down the conclusions.

4. A major disappointment in the study is the failure to identify the neurons in the ARC that provide excitatory input to AVP neurons in response to eating. The authors suggest that the rapid activation of AVP neurons is likely to glutamatergic (lines 492-497) and go on to suggest the Oxtr-expressing neurons are a good candidate. Thus, it is surprising that they tested Pomc- and Agrp-expressing neurons and not the glutamatergic Oxtr-expressing neurons and/or other defined excitatory populations of neurons in the ARC. They should include these experiments if that can be done in a timely manner. They should at least present the eating and drinking data for the indirect experiment presented and discuss the caveats in more detail and tone down their conclusion about the role of the ARC.

*Reviewer #1 (Recommendations for the authors):*

Because of the complexity of this circuitry, a diagram of all the circuitry that they deduce between MnPO/OVLT, SFO and AVP-expressing SON + PVH neurons would be useful for readers unfamiliar with this system.

Line 72, "circuit-mapping techniques" (add hyphen).

Line 81 and elsewhere, Since IRES-Cre was targeted to the mouse Avp locus, it becomes an allele of that locus and should be designated at Avp<IRES-Cre>. Using capital AVP in italics implies that the human gene is targeted. Alternatively, authors can describe the mouse lines without italics.

Line 85, 89, 115 and elsewhere, Adding "in order" to sentences adds little to meaning and could be removed.

Line 117, Authors should only use approved gene names in italics. Vglut2 should be Slc17a6 and Vgat should be Slc32a1. Alternatively, can use the common names without italics. See also comment related to line 81.

Line 157, "suppression… was observed".

Line 339, "genetically accessible and functionally relevant" no hyphens needed when adverbs are used as modifiers. Likewise , line 386: 'temporally precise measurement of circulating AVP' would be better. See also line 412.

*Reviewer #3 (Recommendations for the authors):*

In our experience, some cell types in the LT are insensitive (does not work) to CNO/DREADD in slice. Because Figure 2 assumes functioning DREADD, I would recommend to test it in vitro.

---

## [Author Response]

Essential revisions:1. It is not clearly reported to what extent the chemogenetic silencing of the MnPO/OVLT in the mice used in Figure 2 and 5 reduces the amount of water consumed and how this relates to the dynamics in each animal/trial. This is confounding in two ways. Given that the silencing does not fully block drinking, this implies that the MnPO/OVLT silencing is incomplete (based on Augustine 2018), and thus the negative photometry result in Figure 5 is hard to interpret. Conversely, if silencing reduces drinking partially, which seems likely, then this behavioral change could account for the reduced presystemic inhibition of AVP neurons. It is hard to see how the direct effect of MnPO on AVP neural dynamics could be separated from its effects on behavior in this experiment.

It is a valid concern and we now discuss this point in our manuscript (lines 507-512). However, for the following reasons, we do not believe it is likely that reduced presystemic suppression of AVP neurons is mainly driven by behavioral changes. First, pre-ingestive, cue-induced suppression observed prior to any drinking behavior is completely blocked by silencing of MnPO/OVLT^Vgat^ neurons (Figure 5). Second, MnPO/OVLT neurons provide direct excitatory and inhibitory synaptic inputs to AVP neurons with high probability connections (~100% and ~50%, respectively, Figure 1). Finally, SON-projecting, putative AVP-regulating MnPO/OVLT neurons show water-related presystemic responses that resemble those seen in AVP neurons (Figure 3). Altogether, these factors strongly support the notion that the reduced presystemic suppression of AVP neurons upon silencing of MnPO/OVLT input is primarily caused by direct reduction of the influence of MnPO/OVLT input onto AVP neurons.

2. MnPO neurons are heterogeneous in their dynamics, especially the MnPO-GABA neurons, and for this reason ruling out a possible mechanism based on a photometry trace is challenging. For example, compare the interpretation of the photometry recordings of MnPO-Glp1r neurons in (Augustine, 2018) with the results of single cell imaging of the same neurons in (Zimmerman, 2019). -In Figure 2, DREADD was used to suppress the activity of the LT. However, the virus construct is a general promoter, and no data are provided to demonstrate that CNO/DREADD works in this system or cells. Because there are no behavioral effects of CNO inhibition of SFO or MnPO/OVLT, the authors should confirm the efficiency of the chemogenetic manipulation.

We performed electrophysiology experiments to address reviewers’ concerns regarding effectiveness of CNO/hM4Di. Figure 2—figure supplement 3 shows that CNO/hM4Di effectively silences both MnPO/OVLT and SFO neurons.

3. Although the authors revealed the anatomic sites relevant for different kinds of presystemic regulation of VP neurons, the causal role of specific cell types in these structures that provide this input remains untested/unclear. The manuscript would be significantly more impactful if they tested whether excitatory and inhibitory populations in the MnPO/OVLT indeed mediate the pre- and post-ingestive effects on pre-systemic VP neuronal activity. If these data are in hand or can be generated in a timely manner, they should be included. Otherwise, discuss the caveats and tone down the conclusions.

As suggested by the reviewers, we performed additional experiments to test the involvement of excitatory and inhibitory neurons of the MnPO/OVLT in water-related presystemic regulation of AVP neurons. As now illustrated in Figure 5, effects of silencing excitatory (MnPO/OVLT^Vglut2^) and inhibitory (MnPO/OVLT^Vgat^) neurons were strikingly different. Silencing of MnPO/OVLT^Vglut2^ neurons caused an overall decrease in the presystemic response of SON^AVP^ neurons whereas silencing of MnPO/OVLT^Vgat^ neurons caused a very specific attenuation of pre-ingestive, cue-induced response. These results are consistent with our data in Figure 4 where we demonstrate that MnPO/OVLT^Vglut2^ neurons show both pre- and post-ingestive responses and MnPO/OVLT^Vgat^ neurons show only a pre-ingestive response.

These results directly validate our previous findings and significantly strengthen the conclusion of our paper.

4. A major disappointment in the study is the failure to identify the neurons in the ARC that provide excitatory input to AVP neurons in response to eating. The authors suggest that the rapid activation of AVP neurons is likely to glutamatergic (lines 492-497) and go on to suggest the Oxtr-expressing neurons are a good candidate. Thus, it is surprising that they tested Pomc- and Agrp-expressing neurons and not the glutamatergic Oxtr-expressing neurons and/or other defined excitatory populations of neurons in the ARC. They should include these experiments if that can be done in a timely manner. They should at least present the eating and drinking data for the indirect experiment presented and discuss the caveats in more detail and tone down their conclusion about the role of the ARC.

We agree that this is an extremely interesting question, and we have ambitiously attempted multiple approaches to identify an ARC neuronal population that provide feeding-related presystemic signal to AVP neurons. A main obstacle in targeting glutamatergic ARC population is that we currently do not have a specific genetic marker for these neurons. We could not use Oxtr-cre mice that were used in our original study because we saw cre expression around the SON that prevented us from using Oxtr-cre;AVP-cre mice to selectively target ARC Oxtr and AVP neurons in the same animal. We also attempted using Vglut2-flp;AVP-cre mice but achieving restricted hM4Di expression in the ARC with viral injection was extremely challenging and we decided that the experiment is too inefficient to be completed “in a timely manner”.

As we now present in Figure 7—figure supplement 4, DMH/VMH/ARC silencing did not alter feeding behavior. Therefore, reduced feeding-related presystemic response in AVP neurons are not indirectly caused by slower or less food consumption.

Reviewer #1 (Recommendations for the authors):Because of the complexity of this circuitry, a diagram of all the circuitry that they deduce between MnPO/OVLT, SFO and AVP-expressing SON + PVH neurons would be useful for readers unfamiliar with this system.

We thank the reviewer for suggesting this. We now provide a diagram describing all the circuitry described in our study (Figure 8).

Line 72, "circuit-mapping techniques" (add hyphen).Line 81 and elsewhere, Since IRES-Cre was targeted to the mouse Avp locus, it becomes an allele of that locus and should be designated at Avp<IRES-Cre>. Using capital AVP in italics implies that the human gene is targeted. Alternatively, authors can describe the mouse lines without italics.Line 85, 89, 115 and elsewhere, Adding "in order" to sentences adds little to meaning and could be removed.Line 117, Authors should only use approved gene names in italics. Vglut2 should be Slc17a6 and Vgat should be Slc32a1. Alternatively, can use the common names without italics. See also comment related to line 81.Line 157, "suppression… was observed".Line 339, "genetically accessible and functionally relevant" no hyphens needed when adverbs are used as modifiers. Likewise , line 386: 'temporally precise measurement of circulating AVP' would be better. See also line 412.

We have now revised our manuscript to include reviewer’s suggestions.

Reviewer #3 (Recommendations for the authors):In our experience, some cell types in the LT are insensitive (does not work) to CNO/DREADD in slice. Because Figure 2 assumes functioning DREADD, I would recommend to test it in vitro.

We now provide slice electrophysiology data showing effective silencing of MnPO/OVLT and SFO neurons by CNO/hM4Di (Figure 2—figure supplement 3).